# VeriEquivBench: An Equivalence Score for Ground-Truth-Free Evaluation of Formally Verifiable Code

**Lingfei Zeng**[1]*, **Fengdi Che**[2]*, **Xuhan Huang**[3], **Fei Ye**[4], **Xu Xu**[5],
**Binhang Yuan**[5]†, **Jie Fu**[4]†

[1]Huazhong University of Science and Technology, [2]University of Alberta
[3]The Chinese University of Hong Kong  [4]Shanghai Artificial Intelligence Laboratory
[5]Hong Kong University of Science and Technology

## Abstract

Formal verification is the next frontier for ensuring the correctness of code generated by Large Language Models (LLMs). While methods that co-generate code and formal specifications in formal languages, like Dafny, can, in principle, prove alignment with user intent, progress is bottlenecked by specification quality evaluation. Current benchmarks rely on matching against ground-truth specifications, a manual and expertise-intensive process that has limited existing datasets to a few hundred simple problems and also suffers from a reliability issue. To address this, we introduce VeriEquivBench, a new benchmark with $2,389$ complex algorithmic problems that probe the limitations of current models in both code generation and formal reasoning. Our evaluation framework replaces ground-truth matching with a formally grounded metric, the equivalence score, and rigorously verifies the quality of generated specifications and code. Our results show that generating formally verifiable code remains a profound challenge for state-of-the-art LLMs. This underscores both the difficulty of the task and the need for benchmarks like VeriEquivBench to drive progress toward scalable and reliable coding agents. The datasets are available at https://github.com/PunyGoood/VeriEquivBench.

## 1 Introduction

Large language models (LLMs) already possess substantial capacity for following natural-language instructions and executing a wide range of coding tasks (Li et al., 2022a; Jain et al., 2024; Zhao et al., 2025). At the same time, the correctness of the generated code remains a concern (Cotroneo et al., 2024; Wang et al., 2025b), where functional errors cost users extra effort to debug and also pose significant risks in the safety-critical domain (Dalrymple et al., 2024). A common solution is to evaluate generated code through unit tests (Jimenez et al., 2024; Wang et al., 2025c). However, this process offers no provable guarantee of correctness, as insufficient unit test coverage can fail to detect critical errors (Yu et al., 2025). On the contrary, a verifiable system resolves the issue by co-generating formal specifications and code to formally verify the alignment with the natural language query intention (Sun et al., 2024). Our work focuses on building an end-to-end agent for formal verification, for which we adopt Dafny (Leino, 2010). It is an ideal choice as Dafny's automatic theorem prover (De Moura & Bjørner, 2008) eliminates the need for manual proof writing. Furthermore, its similarity to common languages like Python and C simplifies code transformation.

While several benchmarks (Ye et al., 2025b; Thakur et al., 2025) target at building a reliable reasoning system by formally ensuring the exact code generation (György et al., 2025), their progress is constrained by the reliance on manually-written, ground-truth specifications for

---

*Equal contribution.
†Corresponding author.

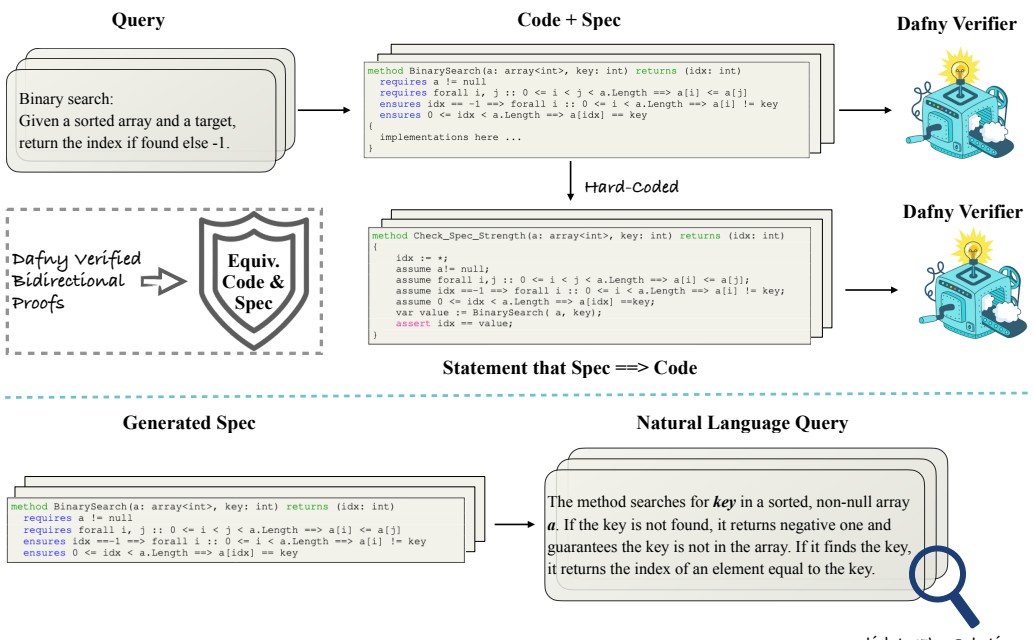

Figure 1: An end-to-end verifiable coding agent first generates both the implementation and its formal specification, and uses the Dafny verifier to prove their mutual equivalence. It then interacts with the user to validate that the formalized intent matches the desired behaviour. The specification-to-code implication can be verified directly by submitting the generated Dafny program to the verifier. To establish the reverse implication, a verification script is constructed from the generated Dafny program to prove that the code does not exhibit behaviours beyond those described by the specification. Together, these two directions of implication establish mutual equivalence, ensuring both the soundness and completeness of the solution.The verified formal specification is subsequently translated back into natural language, enabling the user to confirm that it faithfully captures their original intent.

evaluation. This formal annotation process is incredibly labour-intensive and requires deep expertise (Misu et al., 2024), which sets a barrier to broadly applying formal verification and scaling benchmarks in both size and complexity. As a result, prominent Dafny benchmarks, including DafnySynthesis (Misu et al., 2024) and CloverBench (Sun et al., 2024), contain only 215 simple examples combined, insufficient for evaluating current LLMs' advanced reasoning abilities. Moreover, the reliance on expert annotation is not only a scaling bottleneck; it also leads to a reliability issue. An analysis (Sun et al., 2024) has figured that 10% of expert-written specifications in DafnySynthesis are wrongly claimed as ground-truths, and our own review finds another 18%, containing errors or ambiguities. Such flaws undermine the validity of any benchmark that depends on a ground-truth solution. Therefore, our pipeline requires models to perform automatic formalization, rather than assuming that specifications are given solely for evaluating code quality. This raises a critical question: *How can we reliably evaluate specifications' quality **without** depending on the ground-truth?* To answer this question, we make the following concrete technical contributions:

**Contribution 1.** We propose a novel formally-grounded metric, named the *equivalence score*, that measures the quality of generate specifications as well as the code by verifying their mutual equivalence. The score confirms whether a specification unambiguously describes the code's behaviour by using the Dafny verifier to check for bidirectional implication. This automated process has no false positives, ensuring that only correctly matched code-specification pairs are accepted. In order to validate the alignment with the query intention, we further include a second evaluation step: translating formal specifications back to natural language, as used by Ying et al. and Sun et al.. This allows further interactions with users to fill in ambiguity and clarify their intentions. The complete pipeline is illustrated in Figure 1.

**Contribution 2.** Equipped with our automated evaluation metric, we introduce *VeriEquiv-Bench*, a benchmark of $2,389$ examples with natural language problem descriptions, code and specifications, and additionally, $1,678$ synthetic algorithmic problems. VeriEquiv-Bench significantly expands on prior work in both dataset size and problem complexity, a leap demonstrated by the average Cyclomatic Complexity score, which rises from 2.44 in DafnySynthesis to 5.63. Our benchmark supports evaluating verifiable coding ability given a user query, as well as auxiliary tasks introduced in prior work, such as specification generation from code (Yan et al., 2025) and specification refinement (Loughridge et al., 2025). The core of our dataset is converted from the LeetCode corpus (Xia et al., 2025), a large and community-validated collection of algorithmic problems well-suited for evaluating a model's reasoning abilities. To supplement this data, we also introduce a synthesis pipeline that uses a structured tagging system to generate novel queries by randomly combining tags for different domains, data structures, and algorithms, introduced in Section 2.3. This provides a scalable method for creating large training datasets of new problem descriptions that are fully compatible with our automated evaluation signal.

**Contribution 3.** We conduct a concrete evaluation of state-of-the-art LLMs, where VeriEquivBench serves as a testbed for these models to explore and extend the reasoning abilities on complex problems, beyond human-annotated data (Silver et al., 2021; Ye et al., 2025a; Shojaee et al., 2025). Instead of requiring users to validate the intention satisfaction, we use Claude-4 as a judge (Wang et al., 2025a). Our evaluation highlights the profound difficulty of this task and the effectiveness of our benchmark. The best-performing model, Claude-4-sonnet, which solves 75.81% of the problems on CloverBench, fails on all instances in our dataset, even under the pass@4 evaluation metric. This highlights the limited capacity of current models to carry out reliable formal reasoning on complex algorithmic tasks.

## 2 Benchmark Overview And Construction Pipeline

In this section, we first present aggregate data statistics for VeriEquivBench. Subsequently, we introduce the two curated subsets released with the benchmark: (i) the LeetCode-transformed dataset, and (ii) a tag-composition dataset, called TagComp, the latter being explicitly constructed to evaluate verifiable agents on novel data without contamination (Tu et al., 2024; Riddell et al., 2024).

Each problem in our benchmark provides a comprehensive set of artifacts: a natural language query, implementations in both Python and Dafny, unit tests and two versions of formal specifications: a **strong auto-formalized baseline** explained in Section 2.1 and **a weaker, verifiable but incomplete version** explained in Section 2.2. The strong version of the specification is derived from the natural-language query without omitting any information, so that Claude-4 can generate the correct implementation solely from the specification, without access to the original query. However, this fully detailed version does not pass the verifier. Therefore, we introduce an incomplete but verifiable alternative, referred to as the weak baseline. Additionally, each problem is annotated with metadata, including its difficulty level and descriptive tags for the relevant algorithm, data structure, and domain. Unlike LeetCode, our benchmark uses a more detailed and structured set of tags to categorize problems. This new tagging system is described in Section 2.3 for future query synthesis.

Starting from the original Leetcode split of $2,641$ training and 228 test instances, we first curate $2,174$ cases successfully transformed to Dafny. Then we compose new problems by merging tags, producing $1,893$ additional items; the full tag-composition procedure is described in Section 2.3. For new problems, we ask Claude-4-sonnet to generate pairs of Python code and corresponding unit tests. For only 300 of new problems, Claude-generated code passes at least 85% of their corresponding unit tests, forming the cleaned *TagComp* dataset. Of these, 215 samples clear the weak-baseline pipeline, giving us $2,389$ problems in total that pair natural-language queries with formally annotated code.

Table 1 presents key metrics for our annotated Dafny code, which uses the weaker, verifiably correct specifications. Our problems are significantly more complex than those in Clover-Bench, often involving multiple methods rather than a single one. Furthermore, the corresponding specifications, while incomplete, contain a substantial number of formal clauses.

Table 1: The table overviews several attributes of our annotated code.

| Dataset | Metric | function | method | invariant | ensures | decreases |
|---------|--------|----------|--------|-----------|---------|-----------|
| LeetCode | mean | 0.78 | 1.33 | 5.12 | 1.71 | 0.46 |
| TagComp | mean | 0.96 | 3.18 | 7.34 | 3.14 | 0.70 |

## 2.1 LeetCode Autoformalization

Past formal-language sets such as DafnyBench (Loughridge et al., 2025) are still small and narrow, because hand-written specifications are too costly to scale (Misu et al., 2024). To obtain large, varied training data without extra human cost, we mine the classic Leetcode pool, convert problems to formal specifications, stated in Figure 2 Pipeline 1, while keeping query and specification aligned with two short tightening evaluation protocols (Sun et al., 2024), shown in Figure 2 Pipeline 2.

**Specification Generation**   We feed the problem description to Claude-4-sonnet to obtain an initial Dafny specification, yet even the initial drafts often contain syntax errors. Thus, we revise and resubmit up to ten times until the file has no parse or resolution errors. We find that supplying two simple examples exploits the model's in-context learning (Dong et al., 2023) and sharply lowers the error rate (prompt template in Appendix C).

Furthermore, we constrain the model to generate specifications using only first-order logic, prohibiting recursive or dynamic programming-style definitions. This ensures the specification describes the problem's declarative properties without leaking the implementation.

**Equivalence to NL**   The equivalence check follows the protocol proposed by Clover (2024) and contains two steps: (1) A model (we use Grok4 here) rewrites the description so that it cleanly mirrors the specification, then another model (Claude-4) judges the equivalence between the original description and the rewritten one, yielding a score; (2) The specification alone is translated into Python and executed against the ground-truth LeetCode unit tests. The unit test passing rate is reported in Appendix G.1.

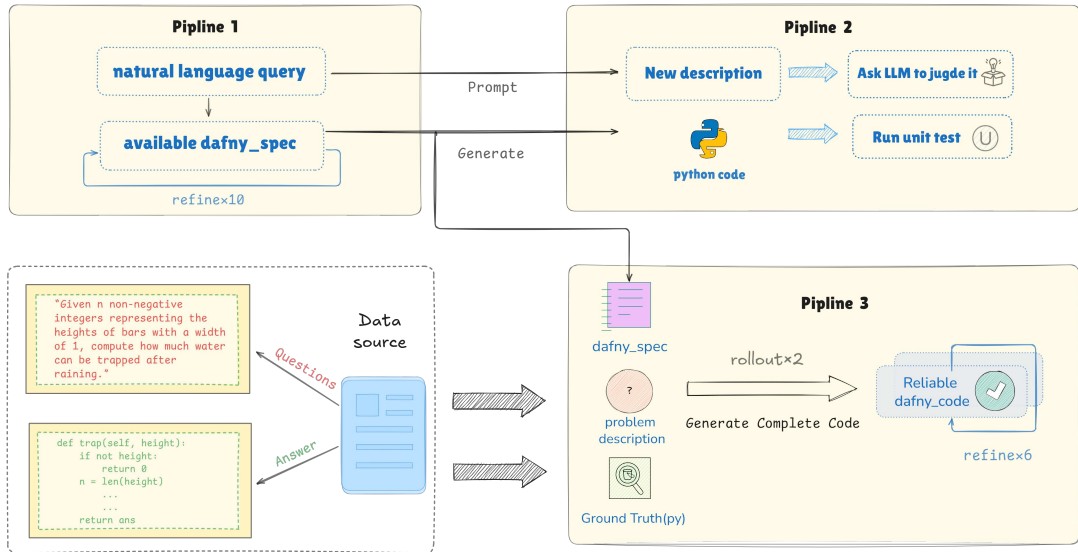

Figure 2: The figure outlines our autoformalization and code generation workflow: Pipeline 1 produces comprehensive and syntax-free specifications; Pipeline 2 checks consistency between the NL query and the specifications; Pipeline 3 emits fully annotated code that passes the verifier.

## 2.2 LeetCode Verifiable Code Generation

Owing to the limited performance of state-of-the-art LLMs on challenging formal-language coding, we adopt the multi-stage pipeline (pipeline 3 of Figure 2): prompted by the previously generated specifications, the problem description, and a reference Python solution, the stronger model (Claude-4) produces annotated Dafny code, while a lighter model (Claude-3.5) then polishes this output, iterating up to six times to eliminate any syntax errors.

In practice, the vast majority of problems converge within three refinement rounds, while a residual subset still fails to yield a well-formed artifact even after the sixth attempt, with the success transformation rate reported in Table 2, and data statistics reported in Table 1.

Table 2: The table shows the number of examples without syntax errors in autoformalization and verifiable examples in code generation.

| Dataset | Spec Autoformalization | | Dafny Code Genetation | |
|---------|----------|----------|----------|----------|
| | Number | Rate (%) | Number | Rate (%) |
| LeetCode | 2584 | 90.1 | 2174 | 75.8 |
| TagComp | 296 | 98.7 | 215 | 71.7 |

## 2.3 Data Synthesis Through Tag Composition

We propose constructing a fine-grained, "template-level" taxonomy to provide an abstract description of algorithmic problems via tags (Wang et al., 2025c). In our system, every task is labelled with three orthogonal categories: **domain**, **data-structure**, and **algorithm** class((Chollet et al., 2025)).

To obtain these labels, we **(i)** harvest a high-quality seed pool from the Luogu online judge (luo, 2025), and **(ii)** manually prune hallucinated or off-topic tags. Our ontology defines over 500 fine-grained tags, offering more than seven times the descriptive granularity of the 69 tags used by LeetCode (see Appendix A for a comparison). The tag set is designed so that, taken together, the tags collectively reflect the complete programming knowledge entailed by each individual problem, while retaining a modest level of abstraction.

The three categories of tags capture complementary aspects of programming knowledge. First, the domain category encompasses the overarching problem space or application context in which an algorithm operates, such as graph theory. Second, the data structure category pertains to the foundational mechanisms for manipulating data that underpin the algorithm's functionality and efficiency, like arrays. Third, the algorithm category refers to the core strategic paradigm employed, such as sorting, focusing on the decision-making logic. These algorithm tags directly shape the overall control flow of a solution, as they orchestrate the program logic and structure.

However, not all problems conform to highly standardized patterns. In contemporary algorithmic competition problems, for instance, many challenges necessitate solvers to discern the underlying mathematical structures, an approach commonly termed "constructive methods". From a coding perspective, these constructive methods typically appear as compact code blocks that rely solely on fundamental loops or arithmetic operations. Consequently, it is difficult to categorize them beyond a general "constructive method" tag. From the problem setter's viewpoint, such problems and their solutions stem from empirically observed mathematical structural properties, which inherently resist exhaustive coverage by conventional tags.

To synthesize novel queries, we select tags in the following workflow: first, we randomly pull 12 tags from each of three pools, and then let Claude-4 pick any 3–8 tags in total. This short list is fed back to the model so that Claude can create one clear algorithm question with roughly 40 unit tests (Xu et al., 2025). Initially, we create approximately 1,900 problems, but only retain the 300 that pass at least 85%of their tests (Xu et al., 2025), and call this clean set, *TagComp*. The detailed pipeline and prompt templates used can be found in Appendix B and Appendix C.

```
method Check_Max_Spec(a: int, b: int) returns (max: int)
    {
      max := *;
      assume max >= a;
      var value := Max( a, b);
      assert value == max;
    }
```

Figure 3: We show an example where the equivalence score proves the given specifications are underspecified for returning the maximum between two integers. The code presents the statement to verify whether the specification implies the code.

## 3 Evaluation Metrics And Tasks

A verifiable coding agent reduces hallucinations and provides trustworthy solutions aligned with users' intentions. As shown in Figure 1, our solution evaluation includes two steps, which are

- verifying the equivalence between generated code and specifications, and
- validating the solution by translating formal specifications back to problem descriptions in natural languages.

To understand the need for verifying the **equivalence** between the code and the specifications, consider a simple binary search algorithm. The goal is to return the index of a *key* in a sorted array *a*, or negative one if the key is not found. A weak but verifiable post-condition might only state that the output, *idx*, is within a valid range: `ensures -1 <= idx < a.Length`. While this specification passes the verifier, it fails to exactly describe the code. This creates a dangerous loophole: an incorrect implementation that doesn't actually find the key could still satisfy this weak condition, and the verifier would not catch the error.

Existing benchmarks do not offer a metric to formally validate the quality of specifications. Without one, there is no way to guarantee that the verified code truly aligns with its intended behaviour. Instead, building up equivalence examines whether the specification is complete without ambiguities. Our equivalence score accomplishes the task by proving the bidirectional implication relationship:

- whether the code falls into the lattices described by the specifications, and
- whether specifications tightly describe the code behaviour for any inputs.

Both proofs can be automatically completed by the Dafny verifier. The first direction can be verified by passing the annotation to the verifier. The second direction requires creating a statement that the specification implies the code for the verifier to check.

Figure 3 presents a counterexample to illustrate how our equivalence score identifies an underspecified function. The `Max` method correctly returns the maximum of two integers *a* and *b*, but its post-condition (`ensures max >= a`) is too weak; it doesn't guarantee that the output is also greater than or equal to *b*. To test if the specification fully implies the code's behaviour, we use the `Check_Max_Spec` method. This method creates an arbitrary value `max`, assumed to satisfy all provided pre-conditions and post-conditions. Our equivalence score then tests the assertion that variables described by the specifications are equal to the method outputs. The Dafny verifier is guaranteed to find this assertion to be false without any false positives. Because the specification is not strong enough to imply the code, this program would not receive an equivalence score.

As mentioned in the introduction, end-to-end formally verifiable code generation is still challenging for current proprietary LLMs. Dafny has its own programming logic, such as claiming the invariance of old elements in arrays to support the proof. Therefore, we re-emphasize the importance of two auxiliary tasks to facilitate understanding of specific

nuances of Dafny, introduced in DafnyBench (Loughridge et al., 2025) and Veri-Code Series I (2025):

- **Verifiable Code Refinement:** Given annotated but unverified Dafny code, the model's goal is to add the necessary intermediate clauses, such as invariants and lemmas, to make the code pass the verifier. Success is determined by successful verification.
- **Code-To-Spec Generation:** Given a Dafny implementation, the model attempts to generate the strongest formal specification. The quality of the output is evaluated by measuring its strength improvement over a baseline, using the spec-superior-score (Yan et al., 2025).

Our two sets of formal specifications map onto these auxiliary tasks. For the Verifiable Code Refinement task, models are challenged to fix our strong auto-formalized specifications. For the Code-to-Specification Generation task, models improve upon our weaker, but already verified, specifications.

## 4 Empirical Evaluation

This section validates the quality of our benchmark and the reliability of our evaluation metric. We then present the performance of several state-of-the-art LLMs on the end-to-end verifiable code generation task, followed by an analysis of our baselines on the two auxiliary tasks.

### 4.1 Quality Metrics

**Specification Quality** Our strong specification baseline, generated via auto-formalization, contains the ground-truth specification for 7.14% of the LeetCode-derived problems and 7.87% of the synthetic TagComp problems, shown in Figure 5. In total, this process yields 161 complex algorithmic data with rigorously verified specifications. This significantly enriches the publicly available dataset of ground-truth specifications.

**Code Transformation Quality** To evaluate the quality of our Python-to-Dafny code transformation, we attempt to validate $1,011$ Dafny programs from the LeetCode set against the corresponding unit tests. Due to the mismatch between Python and Dafny unit test formats, we only successfully execute 648 transformed files. However, the transformation is proven highly reliable, with 81.79% of the translated Dafny programs passing all tests.

**Data Complexity** The average Cyclomatic Complexity (McCabe, 1976) quantitatively manifests the increasing complexity of our data, which counts the number of linearly independent paths in the control flow graph. It is computed using the Radon software package for Python, listed in Table 3.

We list the score for MBPP (Austin et al., 2021), since 50 manually annotated data in DafnySynthesis are based on MBPP-50 and the other 103 are also transformed from it. Thus, the analysis represents a comparison to DafnySynthesis. We skip the analysis of CloverBench due to a lack of Python implementations. Our benchmark's average score of 5.63 is significantly higher than the 2.44 for DafnySynthesis, indicating more complicated control flows. Notably, our synthetically generated data is even slightly more complex than the LeetCode-derived portion, with a score that is 0.25 points higher. This complexity is further validated by a manual rating from Claude-4, which classified the majority of our synthetic problems as either medium or hard.

Table 3: The table compares the code complexity of a previous benchmark and VeriEquiv-Bench, indicating a more intricate control flow of our data.

| Dataset | MBPP-50 | MBPP | LeetCode | TagComp |
|---|---|---|---|---|
| Average Cyclomatic Complexity | 2.44 | 2.78 | 5.38 | 5.63 |

## 4.2 Validation of the Evaluation Metrics

We first validate our equivalence score on 50 expert-written verifiable code provided in DafnySynthesis. CloverBench has reviewed their data and reported that 10% of the data does not give the ground-truth specification. After testing on our evaluation metric, the equivalence score, we figure out another nine examples where the formal specification contains ambiguities or the original code has errors. An example is shown in Figure 4, where the formal specification does not specify the invariance of array length and leaves a logic gap. However, only eight examples out of 14 failures are successfully fixed by us, demonstrating the hardness in manual annotation. All examples with wrongly claimed ground-truth are listed in Appendix G.4 with the issues stated.

```
method SwapFirstAndLast(a: array<int>)
  requires a.Length > 0
  modifies a
  ######## ⇓ The added post-condition
  ensures a.Length == old(a).Length
  ######## ⇑
  ensures a[0] == old(a[a.Length - 1])
  ensures a[a.Length - 1] == old(a[0])
  ensures forall k :: 1 <= k < a.Length - 1 ==> a[k] ==
      old(a[k])
{
    var tmp := a[0];
    a[0] := a[a.Length - 1];
    a[a.Length - 1] := tmp;
}
```

Figure 4: An example of a weak specification in sample #625 that fails equivalence scoring. The formal specification is ambiguous as it omits a post-condition on the invariance of the array's length.

Next, we evaluate all previous benchmarks and observe a serious quality issue in previously provided ground-truth formal specifications, shown in Table 4. It has been discussed that the equivalence check relying on natural language provided in Clover has limitations, and it turns out that a large number of specifications do not establish the equivalence with the code. Meanwhile, DafnyBench is not designed for checking the completeness of specifications and thus, gives the lowest score.

Furthermore, we evaluate Grok-4's translation ability, using Claude-4-sonnet as a judge (Wang et al., 2025a). We test it on our filtered auto-formalized specifications derived from LeetCode and observe a high success rate of 82.98%, validating its functionality to translate.

Table 4: The percentage of data gaining the equivalence score in previous benchmarks.

| Dataset | DafnySynthesis | CloverBench | DafnyBench |
|---|---|---|---|
| Equivalence Score | 76.22% | 61.29% | 43.09% |

## 4.3 Verifiable Code Generation

Figure 5 (b) and (c) present the pass@4 results of three proprietary LLMs on end-to-end formally verifiable code generation, tested on CloverBench and our contamination-free synthetic set, TagComp. We also evaluate three open-source model with complete results presented in Figure 23 in Appendix G.2. On the previous CloverBench benchmark, a capable model like Claude achieves a 75.81% success rate, with most errors stemming from

issues in specification writing rather than fundamental code generation flaws. However, on our more challenging TagComp dataset, this performance collapses dramatically.

A closer look at our benchmark results reveals the challenge of verifiable code generation. In our rigorous two-step evaluation, the equivalence score measures the formal alignment of code and specifications, while the exact matching score further validates against the original natural language intent. While Claude is the most capable model in producing syntactically correct Dafny code, achieving a success rate of 73.79%, all three models struggle substantially to generate verifiable solutions. More critically, they perform poorly at producing mutually equivalent code and specifications that are properly aligned with the user's query intent. GPT achieves the highest mutual-equivalence rate at 2.65%; however, further analysis reveals that these mutually equivalent solutions often provide simplified implementation, hacking the reward and fail to satisfy the intended user requirements. This result underscores the difficulty of formally verifiable code generation on complex algorithmic problems, requiring strong coding and formal reasoning abilities.

Multiple failure cases are provided in Appendix E, where we also illustrate how Claude-4, acting as a judge, identifies incorrect specifications, even in difficult scenarios where only a subset of post-conditions does not fully satisfy the query requirements.

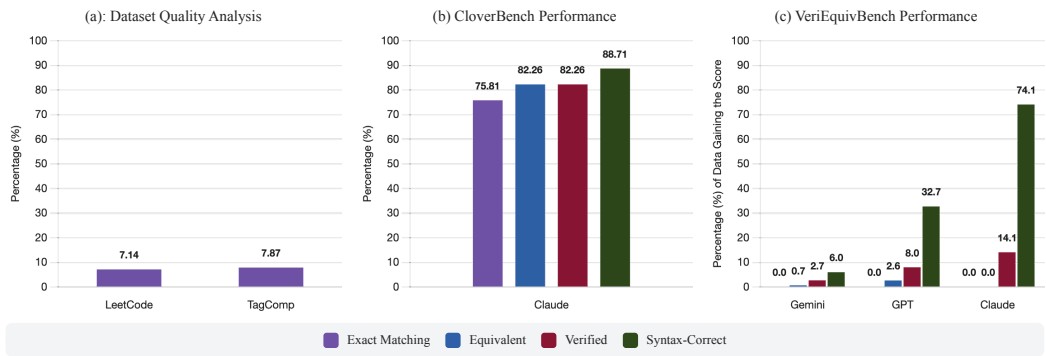

Figure 5: Exact matching score measures the percentage of data passing our two-step evaluation framework, giving solutions aligned with the query intention. Part (a) gives the amount of verified ground-truth solutions in our benchmark. Part (b) shows that the previous CloverBench benchmark is too simple to properly evaluate the advanced reasoning abilities of capable models, as evidenced by a high success rate. Part (c) presents the pass@4 performance of `gemini-2.5-flash`, `gpt-5`, and `claude-4-sonnet` on our end-to-end verifiable code generation task.

## 4.4 Auxiliary Tasks

For the two auxiliary tasks mentioned in Section 3, we provide two RL-trained baselines, with the SFT model provided in Veri-Code Series I (2025). As stated, the verifiable code refinement task uses passing the verifier or not as the reward to infill intermediate clauses, while the spec generation task uses the spec superior score introduced by Yan et al. (2025). Spec superior score measures whether the generation specifications described the code better than our weak baseline. The choice of the RL algorithm and hyperparameters follows their implementation as well. We split our LeetCode transformed data into three parts with 1770 training data, 200 validation data and 204 out-of-domain test data, using tags uncovered by the training data.

Our baseline scores 17.68% for the refinement task and 54% for the spec generation task on the validation set. However, in the spec generation task, almost no data generates a complete specification, resulting in an equivalence score. A possible reason is that the SFT model provided is trained on overly simple problems and does not have enough exploration ability. The training curve and results on the test set are presented in Appendix G.3.

## 5  RELATED WORKS

A central challenge in advancing LLMs is developing metrics that not only assess performance but also provide a clear signal for improvement we desire. **Outcome-based metrics**, such as final-answer accuracy in mathematical reasoning (Cobbe et al., 2021) or pass rates on unit tests in code generation (Austin et al., 2021), are prevalent but limited. They disregard the fidelity of the reasoning process and remain susceptible to false positives, a limitation shared by methods employing external solvers for verification (Huang et al., 2025; Feng et al., 2025).

**Formal verification** offers a more rigorous evaluation alternative, using proof checkers like Dafny (Leino, 2010) or Lean (De Moura et al., 2015) to provide an unambiguous correctness signal without requiring a ground-truth solution. However, in verifiable code generation, this signal is fundamentally unidirectional: it validates that the code satisfies a specification but offers no guarantees about the specification's quality. This vulnerability allows models to pass verification using trivial or flawed specifications (Yan et al., 2025). While Yan et al. (2025) attempt to address this by comparing generated specifications against ground-truth specifications using a partial order, their method remains dependent on the quality and availability of trusted ground-truth. In contrast, our work introduces a formal **equivalence metric** that verifies the bidirectional correspondence between code and specification. This approach ensures the specification fully captures the program's behavior without relying on a ground-truth specification.

The absence of such a metric has hampered the creation of high-quality benchmarks for **autoformalization**. Existing datasets often lack the tripartite alignment of natural language, code, and formal specifications (Lohn & Welleck, 2024; Loughridge et al., 2025; Dougherty & Mehta, 2025; Yan et al., 2025) or are small-scale due to the high cost of manual annotation (Misu et al., 2024; Sun et al., 2024; Miranda et al., 2025; Ye et al., 2025b). Attempts to automate equivalence checking have proven unreliable; for instance, Clover (Sun et al., 2024) relies on LLM-based judgments that suffer from high error rates. Addressing these deficiencies, we present **VeriEquivBench**, a benchmark an order of magnitude larger than prior work. Enabled by our robust equivalence metric, it provides a large-scale, trustworthy resource for developing and evaluating models for verifiable code generation.

## 6  CONCLUSION

In this paper, we confront a foundational challenge hindering the development of reliable verifiable systems: the dependence on small, manually-annotated benchmarks for formal verification. This issue limits the scale and complexity of evaluation and has also introduced a ceiling by human knowledge. Our work breaks the dependency and introduces VeriEquivBench, a large-scale end-to-end formally verifiable code generation benchmark. Our novel automated equivalence score provides a rigorous evaluation signal without any need for human-written, ground-truth specifications. Second, our structured tagging system enables the scalable, automated synthesis of novel and complex problems, directly addressing the data generation bottleneck. By using VeriEquivBench to evaluate state-of-the-art LLMs, we have demonstrated that end-to-end verifiable code generation remains an open challenge, a fact obscured by the inflated success rates on simpler, older benchmarks. Following the recent discussions on self-evolving agents, our benchmark provides a scalable data generation engine and a reliable auto-evaluation metric, setting the groundwork to foster trustworthy AI agents with exact solution generation and sustainably supervise super-intelligence agents.

## 7  REPRODUCIBILITY STATEMENT

The code and our dataset are included in the supplementary material and will be publicly available after the double-blind review process for reproducibility.

## 8 Ethics Statement

This work does not present any foreseeable ethical concerns. The research involves only publicly available datasets and does not use or analyze sensitive or personally identifiable information.

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

# A  Details about algorithm tags

To assemble a suitable tag vocabulary, we first collect high-quality, high-frequency labels from Luogu—a competitive-programming platform with millions of users and an unusually fine-grained tag taxonomy—and treat them as a seed set. For each LeetCode problem, the model is prompted to pick the most relevant domain, data-structure, and algorithm tags from this pool, and is allowed to introduce new tags only when no suitable match exists. All model-selected tags are pooled, automatically partitioned into the three coarse categories, and then manually filtered in a single pass: hallucinated labels are removed, near-duplicates merged, and overly broad or overly narrow tags discarded. The resulting inventory contains over 500 clean triples that serve as the controlled vocabulary for subsequent tag-composition.

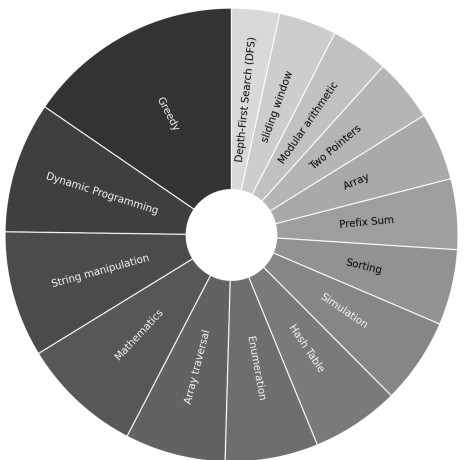

Table 5: Statistics of algorithm tags

| Tag category | Numbers |
|---|---|
| Domain | 53 |
| Data Structure | 68 |
| Algorithm | 480 |

Figure 6: The fifteen most frequently used tags in our dataset.

The complete curated tag set is listed below, grouped under the three top-level categories: domain, data-structure, and algorithm.

Table 6: Domain tags

| Category | Tags |
|---|---|
| **Domain** | Mathematics, Number Theory, Probability Theory, Combinatorial Mathematics, Linear Algebra, Computational Geometry, Plane geometry, Three-dimensional computational geometry, Graph Theory, Simple Graph Theory, Game Theory, Information Theory, Dynamic Connectivity, expectation, Set Cover Problem, allocation problem, Extremum problem, path problem, Chess Board Problem, Stock Problem, Island Problem, Maze Problem, Josephus problem, Frobenius problem, N-Queens Problem, Knight's Tour Problem, Two-dimensional partial order problem, matching problem, Pairing problem, Interval problems, Knapsack problem, Subset Sum Problem, Jump Game, Maximum Subarray Problem, Maximum Subsequence Problem, Largest Rectangle in Histogram, longest chain, Path counting, Path Statistics, Connectivity, Reachability analysis, periodic, Discrete Event Simulation, Time constraint, Permutations and Combinations, Counting Principles, Inclusion-Exclusion Principle, Pigeonhole principle, Catalan number, Stirling numbers of the second kind, Combinatorial counting, Combinatorial Optimization, Mathematical Techniques |

Table 7: Data Structure tags

| Category | Tags |
|---|---|
| **Data Structure** | array, Two-dimensional array, Multidimensional array, sorted array, Circular array, tagged array, Difference Array, rolling array, Linked List, doubly linked list, Circular Linked List, Queue, deque, Priority Queue, Stack, monotonic stack, monotonic queue, tree, undirected tree, unrooted tree, Ring tree, Binary Tree, Complete Binary Tree, Perfect Binary Tree, Balanced Binary Tree, Binary Search Tree, Tree data structure, Trie, Segment Tree, Binary Indexed Tree, Heap, heap - min heap, Huffman tree, Set, Hash Table, Adjacency List, Adjacency Matrix, weight graph, Bipartite graph, Complete graph, Undirected graph, directed graph, Reverse graph, Star graph, Directed Acyclic Graph (DAG), Balanced tree, sparse matrix, Disjoint Set Union (DSU), Red-Black Tree, AVL Tree, B-Tree, B+ Tree, Skip List, Bloom Filter, LRU Cache, Prefix Tree, Suffix Tree, Suffix Array, Cartesian Tree, Splay Tree, Scapegoat Tree, Persistent Data Structure, Linear List, Sparse Table, Mo's Algorithm Structure, Leftist Tree, Fibonacci Heap, Pairing Heap |

Table 8: Algorithm tags

| Category | Tags |
|---|---|
| **Algorithm-1** | Compression algorithm,Dynamic Programming,Dynamic Programming - Linear DP,Dynamic Programming-LIS,Dynamic Programming-Prefix Sum,Dynamic Programming - 0/1 Knapsack,Dynamic Programming - State Compression,Dynamic Programming - Interval DP,Dynamic Programming - 2D DP,Dynamic Programming - Prefix Sum Optimization,Dynamic Programming - Top-Down,Dynamic Programming - Iterative,Dynamic Programming,Compression algorithm,Dynamic Programming,Dynamic Programming - Linear DP,Dynamic Programming-LIS,Dynamic Programming-Prefix Sum,Dynamic Programming - 0/1 Knapsack,Dynamic Programming - State Compression,Dynamic Programming - Interval DP,Dynamic Programming - 2D DP,Dynamic Programming - Prefix Sum Optimization,Dynamic Programming - Top-Down,Dynamic Programming - Iterative,Dynamic Programming, State Compression DP,Dynamic Programming - Mathematical Optimization,Digital DP,Count DP,Tree DP,knapsack DP,State Compression DP,Dynamic Programming (DP),2D DP,Bidirectional DP,Sequence DP,Matrix DP,State Machine DP,Bottom-up Dynamic Programming,Bidirectional BFS,Multi-source BFS,0-1 BFS,Depth-First Search (DFS),Breadth-First Search (BFS),Memoization,State space search,Heuristic search,state search,Grid search,Path Finding,Binary search,Binary Search - Answer,Binary Search - Right Boundary,Binary Search - Left Boundary,Binary Search - Count,Binary Search - Peak Finding,Binary Search - Maximum Value,Binary Search-Prefix Sum,Binary Search - Middle Element,Binary Search - Line Search |

Table 9: Algorithm tags

| Category | Tags |
|---|---|
| **Algorithm-2** | Sorting,Merge sort,Quick Sort,Three-way quicksort,Insertion Sort,Counting Sort,Bucket Sort,Sort-Custom Sort,Sorting - Stable Sort,Sorting - Lexicographical Order,Difference Sorting,multi-condition sorting,Wiggle Sort,in-place sorting,Topological sorting,Quick Select,KMP algorithm,Rabin-Karp algorithm,Manacher's algorithm,suffix array,suffix tree,Z-function,prefix function,string pattern matching,string wildcard matching,backtracking,Enumeration,Binary Enumeration,Subset Enumeration,Combinatorial Enumeration,Two-dimensional enumeration,Simulation,Greedy,Greedy - Interval Operation,Divide and conquer,Divide and Conquer - String Splitting,Divide and Conquer - Closest Pair of Points in a Plane,Central Expansion Method,Staining method,Contribution method,sliding window,Two Pointers,Two Pointers - Sliding Window,Fast and slow pointers,Three Pointers,path compression,Path Tracing,Path reconstruction,Path Planning,Single-Source Shortest Path,Multi-Source Shortest Path,Second shortest circuit,Constrained Shortest Path,shortest path,Heap-optimized Dijkstra,Dijkstra's algorithm,Dijkstra's Algorithm Variant,Bellman-Ford algorithm,Floyd's cycle-finding algorithm,Kruskal's algorithm,Prim's algorithm,Minimum Spanning Tree,Bipartite Matching,Maximum Matching in Bipartite Graphs,Hungarian algorithm,Minimum Cost Maximum Flow,Graham scan,Welzl's algorithm,linear sieve,Euler sieve,Eratosthenes sieve,Prime Sieve, Euclidean algorithm,Bézout's identity,Bézout's theorem,Greatest Common Divisor (GCD),Least Common Multiple (LCM),Prime Number Check |

Table 10: Algorithm tags

| Category | Tags |
|---|---|
| **Algorithm-3** | Euclidean algorithm, Bézout's identity, Bézout's theorem,Greatest Common Divisor(GCD),Least Common Multiple(LCM),Prime Number Check,Prime factorization, Factorization,Integer factorization,Cantor expansion,Fast exponentiation,Matrix Fast Exponentiation,Matrix multiplication,matrix rotation,matrix transposition,Matrix operations,rotation matrix, flood fill algorithm,A* algorithm,Tarjan's algorithm,Morris traversal,Preorder Traversal, Inorder Traversal,Postorder traversal,Level order traversal,Level Order Traversal,Reverse inorder traversal,zigzag traversal,spiral, traversal,Vertical traversal,Vertical Order Traversal,Boundary traversal,Diagonal Traversal,2D matrix traversal,Traversal of 2D Array, Graph traversal,Linked list traversal,Tree traversal,Directional traversal,Bidirectional traversal,reverse traversal,Reverse traversal,One-pass traversal,Path Validation,Path counting,Path Statistics,Path Construction,lexicographical comparison,Lexicographically smallest path,Maximum Value Search,Maximum Value Maintenance,Range Maximum,Maximum Column Value,prefix maximum,suffix minimum,suffix product,prefix product,Prefix Sum,Prefix Sum - Difference,Prefix Sum - Modular Arithmetic,Prefix Sum - Binary Search Optimization,2D prefix sum,suffix sum,partial sum, subarray sum, submatrix sum, Area Sum,Area Calculation,ASCII code manipulation,Character Mapping,Character Count,character frequency,Digital encoding,Digital Parsing,Data Extraction,Number Reversal,Integer Reversal,Integer Square Root,Integer Division,Fraction Addition and Subtraction,Fractional Arithmetic,Fraction simplification,Score Calculation,percentile,Circular shift,Loop Detection,Ring Detection,Periodic Assessment,Bracket Matching,Isomorphic Strings,String comparison,String Case Conversion,String concatenation,string concatenation,String manipulation,String search,string matching,String-Substring Comparison,string-replacement,String replacement,String trimming,string slicing,string splitting,String compression,String decoding,string parsing,string continuity,substring matching,prefix matching,Prefix Check,Longest Common Prefix,Longest Common Suffix,Longest Common Subsequence,Longest Common Subarray,Longest Repeating Substring,Longest Palindromic Subsequence,Longest Non-decreasing Subarray,Longest Consecutive Sequence,longest consecutive characters,Word Chain,Zigzag Conversion,palindrome,Expression parsing,Expression Evaluation,Reverse Polish Notation,Postfix expression,Operator precedence,Lexical Analysis,parsing,Serialization,Deserialization,Encoding,decoding,Run-length encoding,Set Operations,Set Intersection,Bitwise operation,Bitwise operation optimization,Bitwise Operations - State Compression,bitmask,Bitwise OR, AND operation,XOR,binary,Binary Addition,binary splitting,Binary counting,bit count,Hamming distance,Two's complement,Modular arithmetic,modulo 3 operation,Congruence,Congruence theorem,divisible,Divisibility property,divisor,perfect square,square number,Perfect number,Ugly number,trailing zeros,digit separation,Digital Processing,Digital Sum,Gray code,Permutation, Next Permutation,Arrangement,Permutation ring,Cyclic permutation,Pascal's triangle,Fermat's theorem on sums of two, squares,Pythagorean theorem,Triangle inequality,absolute value,absolute value inequality,Big Integer Addition,High precision |

Table 11: Algorithm tags

| Category | Tags |
| --- | --- |
| **Algorithm-4** | Floating-point processing,Floating-point comparison,floating-point precision,Linear equation,polynomial,Complex Number Operations,Rational number representation,recurring decimal,factorial,Sum of Squares,Sum,Summation formula,arithmetic sequence,Arithmetic sequence summation,path sum,Maximum Sum Path,Maximum spacing,Neighbor Count,Adjacent elements,Adjacent Element Difference,Global Inversion,Local inversion pairs,Inversion pair,anagram,vowel substitution,coordinate,2D coordinates,coordinate system,coordinate comparison,coordinate translation,coordinate compression,2D offset,2D plane,3D space,collinear points,Collinearity detection,convex hull,minimum bounding rectangle,Triangle Area,Rectangle Area Calculation,Overlapping Area Calculation,Rectangle Intersection,Circle-Rectangle Intersection Detection,Minimum Enclosing Circle,Spatial segmentation,2D cutting,Spatial optimization,Space complexity optimization,Constant space complexity,Linear space complexity,Time complexity analysis,Linear time complexity,Linear scan,Pruning,Preprocessing,preprocessing,Offline processing,Dynamic update,Dynamic Maintenance,Dynamic Maintenance Interval,Dynamic Range Maintenance,Single-point modification,Range query,Interval computation,Interval Statistics,Range update,Interval Merging,Interval coverage,Interval Scheduling,Range extrema,Path Intersection Detection,Distance calculation,Euclidean distance,Manhattan distance,Chebyshev distance,projection,cross product,Polar sorting,construct,Binary Construction,Tree Construction,Tree Reconstruction,Sequence Reconstruction,Constructing the answer in reverse order,reverse,Reverse Linked List,Linked List Reversal,String Reversal,Array Rearrangement,Linked List Reordering,Node switching,Segmentation,Split Array,split string,Split and Merge,Convert 1D Array to 2D Array,matrix,2D matrix,sparse matrix,ordered matrix,Rectangle Coverage,Adjacency Matrix,Tree deletion operation,Tree depth,Tree Centroid,Tree Diameter,subtree,Subtree Sum,leaf node,intermediate node,dummy node,sentinel node,Middle of the Linked List,indegree,indegree and outdegree,degree,degree sequence,Monotonicity,Monotonicity Check,monotonic array,Decision Monotonicity,Symmetric,Boolean operations,Logical Operations,Conditional statement,Filter Criteria,Polarity,Parity Check,Boundary check,Boundary handling,Edge case handling,Status Check,Status Log,State transition,State Machine,Finite State Automaton,Priority, handling,Query Processing,Path processing,Overflow handling,Carry handling,Recursion,recursive,Inductive method,derivation,traverse,Array traversal,Grid traversal,directional search,State compression,Handling Duplicate Elements,deduplication,Enumeration optimization,Sequence comparison,comparison function,Comparator,Regular Expression,Pointer manipulation,Method chaining,Swap operation,Displacement operation,Row and Column Operations,product,Multiplication Principle,Exponentiation,Base,Base Conversion,Clock issues,loop section,IP address,reordering,Partial Ordering,Equation Solving,Randomization,reverse thinking,Horse Racing Strategy,Connected component,Connected Component,Strongly Connected Component,Lowest Common Ancestor (LCA),Eulerian circuit,Hamiltonian path |

# B PIPELINE OF TAG COMPOSITION

Figure 7 illustrates our pipeline for generating new programming problems through tag composition. The process begins by creating a candidate pool of 36 tags, randomly selecting 12 from each of our three categories: domain, algorithm, and data structure. This pool is provided to an LLM, which is prompted to select a coherent subset of three to eight tags that form a promising basis for a new problem. Using this selected combination, we then instruct the LLM to generate a complete task, comprising a problem description, corresponding unit tests, and a Python solution. As a final quality control step, we filter these generations by executing the unit tests. We retain only those instances where the generated Python code passes all tests, ultimately yielding a dataset of 300 validated programs.

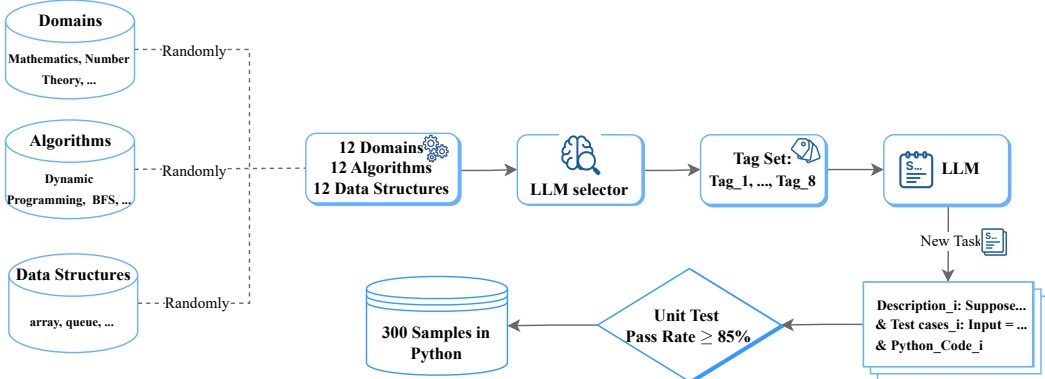

Figure 7: The pipeline for the tag compositon process.

## C    PROMPT TEMPLATES

### C.1    NOVEL TAG COMBINATION

---

[Task]
You have three categories of tags: domain, algorithm, and data_structure, each containing 12 tags. Your task is to select a combination of 3–8 tags from these categories to form a coherent programming problem. The problem should have a specified difficulty level: easy, medium, or hard. Ensure the selected tags are compatible and can logically form a single problem. Provide the chosen tags, the difficulty level.

[Requirements]

1.The task is clearly defined, specifying the need to select 3–8 tags from three categories (domain, algorithm, data_structure) to form a coherent programming problem with a specified difficulty level.
2.Requirements outline the tag selection process, ensuring compatibility and a reasonable tag collection, the need for a difficulty level.
3.The selected tags must be compatible and form a reasonable tag collection that results in a practical and solvable programming problem.
4.The problem must be assigned one of three difficulty levels: easy, medium, or hard, reflecting the complexity of the problem based on the selected tags.

[Domain tags]
{{ domain_tags }}

[Algorithm tags]
{{ algo_tags }}

[Data Structure tags]
{{ data_tags }}

Output Format
This is the ouput format,You must respond in this specified output format:

```
{
    "all_tags": "Graph Theory, Depth-First Search, Union-Find, Graph,
        Disjoint Set",
    "Domain": "Graph Theory",
    "Algorithm": "Depth-First Search, Union-Find",
    "Data_Structure":"Graph, Disjoint Set" ,
    "Difficulty Level": "medium",
}
```

<|Problem End|>

---

Figure 8: The prompt is for selecting useful tags. We feed the model the 36 real tags from 3 categories randomly that will later drive new-problem generation, it returns the 3–8 tags that form the most promising combination.

## C.2 NOVEL PROBLEM SYNTHESIS

[Task]

You are an expert algorithm problem creator. Your task is to create an easy or medium difficulty ranking original coding problem using the given algorithm tags.Analyze the given tags to generate a new problem.The problem should be completely original coding problem that is NOT from any existing platforms (LeetCode, Codeforces, etc.) or textbooks.

[Requirements]

1. Create a truly novel problem scenario with constraints
2. Combine the given tags in innovative ways
3. Ensure the problem is solvable but challenging
4. Provide a clear problem statement, examples, and constraints
5. Rate the difficulty(easy, medium, hard) appropriately

[Algorithm tags]
tags

Output Format

This is the output format. You must respond in this specified output format:
<|Problem Begin|>
problem
<|Problem End|>

Figure 9: The prompt uses the previously obtained real tags to generate a brand-new problem.

## C.3 SPEC-TO-NL

[Prompt]

Can you think of a minimal code implementation satisfying the specification? For example, if the spec just ensures true, then any code can work. If the specification ensures return values within a range, then assigning any value within the range can work. Please think of the minimum code implementation and then come up a problem description this minimal code satisfies.
Below are the specifications:

Figure 10: The prompt asks the model to read the supplied Dafny specification and produce a concise summary that fully describes the coding problem it defines.

## C.4 LLM-As-A-Judge

[Prompt]

You are an expert in analyzing algorithm problem descriptions. You need to carefully analyze the equivalence of two algorithm descriptions based on the following dimensions:
1. Core Problem Equivalence: - Is the essence of the problem identical? - Are the solution objectives consistent?
2. Constraint Comparison: - Input constraints - Boundary case handling - Special case requirements
3. Complexity Requirements: - Time complexity requirements - Space complexity requirements
4. Detail Completeness: - Information loss check - Additional information analysis
Please provide an equivalence score from 0-100 and give a detailed analysis of your reasoning.
Please analyze the equivalence between the following two algorithm descriptions:
Original Description:
New Description:
Please analyze according to the dimensions above and provide a score with detailed explanation.Only put the score in a code block surrounded by triple backticks ("')"""

Figure 11: The prompt instructs the model to determine whether the two given programming problems are semantically equivalent.

## C.5 NL Query To Verifiable Code

> [Prompt]
>
> You will get a problem description. Your task is to give a fully verified Dafny program. Refer to the Dafny examples as guidance:
> Fewshot Examples:
> Problem description:
> Please write the Dafny code that implements the functionality while ensuring:
>
> 1. Reference the Python implementation for algorithmic insights;
>
> 2. Add appropriate loop invariants with brief explanations;
>
> 3. Ensure full verification - your code must pass the Dafny verifier.
>
> Output the complete Dafny program, including both the specification and implementation.

Figure 12: The prompt turns a natural-language query into a fully formal, verifiable specification together with correct-by-construction code.

## D  Model Argument Settings

Throughout all experiments, we retained the default values for every hyperparameter except temperature and top-p. To balance creativity with reliability, we employed a two-level sampling strategy: during the initial specification-generation stage shown in Pipeline 1 in Figure 2, temperature was set to 0.7 and top-p to 0.9 to encourage diversity for generating high-quality formal specifications equivalent to the NL query(Li et al., 2022b).

In all other phases, including annotated code generation in Pipeline 3 in Figure 2 and model evaluation, temperature was reduced to 0.5 and top-p to 0.8 to promote deterministic and consistent outputs.

The prompts are provided in Section C.

For model evaluation, the coding agent is provided with the problem in natural language and is asked to generate four rollouts of annotated Dafny code. The equivalence score is then evaluated for each rollout. Next, those rollouts that gain the equivalent score are passed to Grok-4 to translate specifications back into NL. Finally, Claude-4 judges the equivalence between the translated new description and the original query.

# E  Qualitative Analysis

## E.1  Examples of Verification Failures

Most failures come from unprovable clauses, including missing intermediate proofs or unspecified conditions, as shown in Figure 13. However, to be noticed, Dafny has strict requirements for writing specifications in order to pass the verifier. We have provided two examples, whose specifications are correct and follow the syntax rules, in Figure 14 and 15. However, the Dafny verifier requires re-expressing the code in a different way to pass the verifier.

---

[Example 1]

```
decreases grid.Length0 * grid.Length1 -CountVisitedLandCells(grid
    , visited)
```

Error: *decreases* expression might not decrease.
Error: *decreases* expression must be bounded below by 0 at the end of the loop iteration.

[Example 2]

```
totalCost := totalCost + energyCosts[reachable[i]];
```

Error: index out of range.

---

Figure 13: We provide two examples which cannot pass the verifier with missing intermediate clauses.

```
while i <= |text| - |pattern|
invariant 0 <= i <= |text| - |pattern| + 1
invariant forall j :: 0 <= j < i ==> text[j..j+|pattern|] != pattern
{
    if i + |pattern| <= |text| && text[i..i+|pattern|] == pattern {
        return true;
    }
    i := i + 1;
}
```

Figure 14: An example of Claude-generated code and specifications that cannot be verified by the Dafny verifier, although all specifications are correctly written.

```
method MountainPathNavigation(elevations: seq<int>, queries: seq<int
    >) returns (results: seq<int>)
requires |elevations| > 0
requires forall i :: 0 <= i < |elevations| - 1 ==> elevations[i] <=
    elevations[i+1]  // non-decreasing
requires forall k :: k in queries ==> 0 <= k < |elevations|
ensures |results| == |queries|
ensures forall i :: 0 <= i < |results| ==>
    (results[i] == -1 <==> (forall pos ::
        queries[i] <= pos < |elevations|  ==> !HasDescendingPath(
        elevations, pos))) &&
    (results[i] != -1 ==> queries[i] <= results[i] < |elevations|  &&
        HasDescendingPath(elevations, results[i]) &&
     (forall pos :: queries[i] <= pos < results[i]  ==> !
        HasDescendingPath(elevations, pos)))
{
    // First, precompute all valid positions
    var validPositions := FindValidPositions(elevations);

    // Answer each query using binary search
    results := [];
    for i := 0 to |queries|
        invariant |results| == i
        invariant forall j :: 0 <= j < i ==>
            (results[j] == -1 <==> (forall pos ::
                queries[j] <= pos < |elevations|  ==> !HasDescendingPath(
                elevations, pos))) &&
            (results[j] != -1 ==>
                queries[j] <= results[j] < |elevations|  &&
                HasDescendingPath(elevations, results[j]) &&
             (forall pos :: queries[j] <= pos < results[j]  ==> !
                HasDescendingPath(elevations, pos)))
    {
        var result := BinarySearchLeftmost(validPositions, queries[i
            ]);
        results := results + [result];
    }
}

predicate HasDescendingPath(elevations: seq<int>, start: int)
    requires 0 <= start < |elevations|
{
    exists end :: start < end < |elevations| &&
        (forall k :: start <= k < end ==> elevations[k] > elevations[
            k+1])
}
```

Figure 15: An example of Claude-generated code and specifications that cannot be verified by the Dafny verifier. Although all specifications are correctly written, it does not follow the Dafny grammar.

The example in Figure 15 shows the limitation of the Dafny verifier. Consider the following two preconditions:

$$\text{requires forall } k :: k \text{ in queries} ==> 0 <= k < |\text{elevations}|,$$

and

$$\text{requires forall } i :: 0 <= i < |\text{queries}| ==> 0 <= \text{queries}[i] < |\text{elevations}|.$$

Although two preconditions express the same semantic content, the first condition generated by Claude causes verification errors for the highlighted part in Figure 15; the range of each element in queries cannot be proven. However, switching to the second precondition solves the issue because the second precondition limits the range of each position needed for the verifier.

### E.2 An Example of Ambiguous Specifications

In this subsection, we present an example whose specifications are too weak to describe the code behaviour and cannot pass our whole pipeline without alignment with the original user intention.

**Problem Description**  You are a security consultant for a museum that has a complex layout of interconnected rooms. The museum has motion sensors that detect when visitors move between rooms, and you need to validate if a recorded sequence of room visits represents a valid path through the museum. The museum layout is represented as an adjacency matrix where 1 indicates a direct connection between two rooms, and 0 indicates no direct connection. Additionally, the museum has special 'checkpoint rooms' that visitors must pass through in a specific order when moving between certain sections. Your task is to validate a given path and determine if it's physically possible given the room connections, and also verify that all checkpoint rooms are visited in the correct sequence.

First line of the input contains integer n (number of rooms). Next n lines contain the adjacency matrix ($n \times n$) representing room connections. Next line contains integer $k$ (number of checkpoint rooms). Next line contains $k$ integers representing the required order of checkpoint rooms. Finally, the last line contains the path as a sequence of room numbers to validate.",

Return 'VALID' if the path is valid (all consecutive rooms are connected and checkpoints are visited in order), 'INVALID_CONNECTION' if there's an invalid room transition, 'INVALID_CHECKPOINT' if checkpoints are not visited in the required order, or 'MISSING_CHECKPOINT' if not all checkpoints are visited.

**Ambiguous Spec Generated By Claude**  The specification is shown in Figure 16.

```
method ValidateMuseumPath(n: int, adjacency_matrix: array2<int>,
    checkpoints: array<int>, path: array<int>)
returns (result: string)
requires n >= 0
requires adjacency_matrix.Length0 == n &&
    adjacency_matrix.Length1 == n
requires forall i, j :: 0 <= i < n && 0 <= j < n ==>
    adjacency_matrix[i, j] == 0 || adjacency_matrix[i, j] == 1
requires forall i :: 0 <= i < checkpoints.Length ==>
    0 <= checkpoints[i] < n
ensures result in {"VALID", "INVALID_CONNECTION", "INVALID_CHECKPOINT
    ", "MISSING_CHECKPOINT"}
```

Figure 16: Here is an example whose postconditions are too loose to describe the code behaviour.

```
method ValidateMuseumPath(n: int, adjacencyMatrix: array2<int>, k:
    int, checkpoints: array<int>, path: array<int> ) returns (result:
     int)
requires 3 <= n <= 100
requires 1 <= k <= n
requires k <= path.Length <= 1000

requires adjacencyMatrix.Length0 == n && adjacencyMatrix.Length1 == n
requires checkpoints.Length == k
requires path.Length >= 1
requires forall i, j :: 0 <= i < n && 0 <= j < n ==>
    adjacencyMatrix[i, j] == 0 || adjacencyMatrix[i, j] == 1
requires forall i :: 0 <= i < k ==> 0 <= checkpoints[i] < n
requires forall i :: 0 <= i < path.Length ==> 0 <= path[i] < n
requires forall i, j :: 0 <= i < j < k ==>
    checkpoints[i] != checkpoints[j]

ensures 0 <= result <= 3

ensures result == 0 <==> (
    (forall i :: 0 <= i < path.Length - 1 ==>
        adjacencyMatrix[path[i], path[i+1]] == 1) &&
    (forall cp :: 0 <= cp < k ==> exists i ::
        0 <= i < path.Length && path[i] == checkpoints[cp]) &&
    (forall cp1, cp2 :: 0 <= cp1 < cp2 < k ==>
        exists i1, i2 :: 0 <= i1 < i2 < path.Length &&
        path[i1] == checkpoints[cp1] && path[i2] == checkpoints[cp2])
)

ensures result == 1 <==> (
    exists i :: 0 <= i < path.Length - 1 &&
        adjacencyMatrix[path[i], path[i+1]] == 0
)

ensures result == 3 <==> (
    (forall i :: 0 <= i < path.Length - 1 ==>
        adjacencyMatrix[path[i], path[i+1]] == 1) &&
    (exists cp :: 0 <= cp < k && forall i :: 0 <= i < path.Length ==>
         path[i] != checkpoints[cp])
)
ensures result == 2 <==> (
    (forall i :: 0 <= i < path.Length - 1 ==>
        adjacencyMatrix[path[i], path[i+1]] == 1) &&
    (forall cp :: 0 <= cp < k ==> exists i :: 0 <= i < path.Length &&
         path[i] == checkpoints[cp]) &&
    (exists cp1, cp2 :: 0 <= cp1 < cp2 < k &&
        forall i1, i2 :: (0 <= i1 < i2 < path.Length &&
        path[i1] == checkpoints[cp1] &&
            path[i2] == checkpoints[cp2]) ==> false)
)
ensures result == 1 ==> (
    exists i :: 0 <= i < path.Length - 1 &&
        adjacencyMatrix[path[i], path[i+1]] == 0
)
ensures (forall i :: 0 <= i < path.Length - 1 ==>
    adjacencyMatrix[path[i], path[i+1]] == 1) ==>
    (result == 0 || result == 2 || result == 3)
```

Figure 17: Here is the strong specification generated using our pipeline. Our pipeline translates natural language to specifications first and then reinserts them into the generated code.

**Translation by Grok-4**  This method addresses a simplified museum path validation problem, where the museum is modeled as a graph with n rooms (nodes) connected via a binary adjacency matrix, a set of checkpoint rooms that might need to be visited, and a proposed path sequence through the rooms. However, due to a policy of unconditional approval (perhaps for a prototype or always-valid scenario), the function always deems the path valid and returns "VALID", without checking connections, checkpoints, or path validity. It assumes all inputs are well-formed (valid matrix size, binary entries, and checkpoint ranges) and provides no detailed error reporting, effectively serving as a placeholder validator that optimistically approves any tour route in reinforcement learning datasets for graph traversal experiments. If needed, it could return other statuses like "INVALID_CONNECTION" in variants, but here it guarantees approval to satisfy minimal validation specs.

**Equivalence Judgment by Claude-4**  Final Equivalence Score: 0.

These two descriptions have zero equivalence. They describe completely unrelated problems: Different domains: Mountain elevation analysis vs. graph-based museum navigation; Different data structures: 1D sorted array vs. 2D adjacency matrix with paths; Different operations: Search for valid positions vs. unconditional approval; Different computational requirements: Actual algorithm vs. placeholder function; Different I/O formats: Numerical queries/positions vs. validation strings.

The only superficial similarity is that both involve some form of path or traversal concept, but even this is fundamentally different (descending sequences in an array vs. connected paths in a graph). There is no meaningful relationship between these two algorithm descriptions.

**Specification from Our Pipeline**  Figure 17 presents the generated specifications using our pipeline for the same question. Our pipeline translates natural language to specifications first and then reinserts them into the generated code. It leads to a noticeable improvement in quality compared to code and specification generation being done simultaneously. This reveals an interesting weakness of current models: models can separately generate specifications and code, but fail to do them together.

### E.3  A Hard Example for Spec-to-NL Translation

In the above example, we remove any post-conditions on verifying the connectivity of the path in the specification to create a hard test example for Spec-to-NL translation. This difference is successfully captured when translating the specifications back to natural language and observed by the Judge model.

**Translation by Grok-4**  Problem: Subsequence Order Validator

Given an integer n (unused, for compatibility), an $n \times n$ adjacency matrix (unused, for compatibility), an integer $k$ ($1 <= k <= n$), an array of $k$ unique checkpoints (room indices from 0 to n-1), and an array representing a path (sequence of room indices from 0 to n-1, length at least $k$), validate whether the path contains the exact sequence of checkpoints as a (possibly non-consecutive) subsequence. Return 0 if it does (VALID), 2 if all checkpoints appear but at least one pair has all occurrences of the earlier checkpoint after all occurrences of the later one (INVALID_CHECKPOINT order violation), 3 if at least one checkpoint does not appear in the path (MISSING_CHECKPOINT), and never return 1 (INVALID_CONNECTION, as connections are not checked). This can be used to check if a sequence of events includes required milestones in a specified relative order, ignoring any graph structure.

**Equivalence Judement by Claude-4**  Equivalence Score: 30.

Why this low score (30/100): Missing core functionality (-40 points): The new description explicitly removes graph connectivity validation, which is a major component of the original problem. This is not a minor detail - it's half the problem.

Different output behavior (-20 points): Cannot return INVALID_CONNECTION, leading to incorrect results for inputs with connection violations.

Explicit admission of incompleteness (-10 points): The new description acknowledges inputs are "unused, for compatibility," indicating it's deliberately solving a subset problem.

Why not lower (0-30): Shared checkpoint validation (+20 points): The checkpoint ordering logic appears identical between both descriptions.

Same input format (+5 points): Both accept the same input structure (even if new ignores parts).

Overlapping outputs (+5 points): Three of four output codes have the same meaning.

### E.4 An Example with Tight Specifications but Mismatched Code

We also provide another example of wrongly generated code. Although the generated specifications accurately describe the code, the alignment between the code and the natural-language query is not guaranteed and requires a second-step verification. Our pipeline successfully captures the mismatch, which cannot be achieved by any other code generation benchmark. Considering that this incorrect code passes 85% of co-generated unit tests, we doubt the reliability of using LLM-generated unit tests to verify code.

**Problem Description**  A chess tournament organizer needs to create a bracket system for N players. The tournament follows a complete binary tree structure where each leaf represents a player, and each internal node represents a match between winners of its subtrees. The organizer has a sorted array of player ratings and wants to generate all possible valid tournament brackets using backtracking. A valid bracket must satisfy: 1) Each player appears exactly once as a leaf, 2) The tournament tree is complete (all levels filled except possibly the last), 3) Players are assigned to positions such that the bracket can accommodate exactly N players. You need to find the number of distinct ways to arrange players in valid tournament brackets.

**Incorrect Code**  The code, with its corresponding specification, is shown in Figure 18.

**Translation by Grok-4**  Problem: Tournament Outcome Counter You are organizing a single-elimination tournament where players are eliminated one by one until only the champion remains. Given a list of players (with 1 to 16 players), write a function tournament_bracket_count that calculates the total number of possible outcomes for the tournament, assuming a fixed bracket structure. Each game has exactly two possible results (one player wins, the other loses), and a tournament with n players requires exactly n-1 games. Therefore, the number of possible outcomes is $2^{(n-1)}$.

Input: A list of players (e.g., a sequence of integers representing player IDs).

Constraints: $1 <=$ number of players $<= 16$.

Output: An integer representing the number of possible tournament outcomes.

**Equivalence Judgment by Claude-4**  Equivalence Score: 5.

The score of 5/100 reflects that while the new description uses similar terminology (peaks, revenues, maximum sum), it describes implementations that: Do not solve the original problem - returning constant 1 or the first element is not finding a maximum mountain-pattern subsequence; Ignore all constraints - no mountain pattern validation whatsoever; Have no algorithmic content - $O(1)$ stub vs.$O(n \times m)$ algorithm; Are explicitly described as placeholders - the text itself admits these are "simplified" versions that "ignore the actual revenue data".

The 5 points are awarded only for: Using the same domain context (ski resort/mountain revenues); Sharing vocabulary (peaks, revenues, maximum); Acknowledging the existence of input data structures.

```
function exponential(base: int, exp: nat): int
  decreases exp;
{
  if exp == 0 then 1
  else base * exponential(base, exp-1)
}

method TournamentBracketCount(players: seq<int>) returns (count: int)
  requires 1 <= |players| <= 16
  ensures count == exponential(2, |players|-1)
{
  var n: int := |players|;
  count := 1;
  var i: int := 1;
  if n == 1 {
      return;
  }
  while i < n
    invariant 1 <= i <= n
    invariant count == exponential(2, i-1)
    decreases n - i
  {
      count := count * 2;
      i := i + 1;
  }
}
```

Figure 18: Here is an example whose generated code does not satisfy the user request but has tight specifications generated along with it.

This is essentially not an equivalent algorithm but rather a pair of stub functions that might be used as placeholders before implementing the actual solution. The new description reads like documentation for incomplete/prototype code rather than a working implementation of the mountain revenue problem.

# F    Examples of Synthesis Problems

```
{
    "title": "Magical Crystal Collection",
    "difficulty": "medium",
    "description": "You are an adventurer exploring a magical cave with n
        crystals, each having a power value. The cave has a special
        property: you can only collect crystals whose total power equals
        exactly one of k magical target values. However, there's a twist
        - the cave becomes unstable after collecting crystals, so you
        want to find the target value that can be achieved using the
        minimum number of crystals. If multiple targets require the same
        minimum number of crystals, return the smallest target value. You
         need to efficiently determine which magical target is achievable
         and requires the fewest crystals.",
    "input_format": "First line contains two integers n and k. Second
        line contains n integers representing crystal power values. Third
         line contains k integers representing the magical target values
        .",
    "output_format": "Return the magical target value that can be
        achieved with minimum crystals, or -1 if no target is achievable
        .",
    "constraints": [
        "1 <= n <= 20",
        "1 <= k <= 1000",
        "1 <= crystal power <= 1000",
        "1 <= target values <= 20000",
        "All crystal powers are positive integers",
        "All target values are distinct"
    ],
    "examples": [
        {
            "input": "4 3\n2 3 5 7\n10 12 17\n",
            "output": "10",
            "explanation": "Target 10 can be achieved with crystals [3,7]
                 (2 crystals), target 12 can be achieved with [2,3,7] (3
                crystals), target 17 can be achieved with [2,3,5,7] (4
                crystals). Minimum is 2 crystals for target 10."
        },
        {
            "input": "3 2\n1 4 6\n8 11\n",
            "output": "11",
            "explanation": "Target 8 cannot be achieved with any subset.
                Target 11 can be achieved with [1,4,6] (3 crystals). So
                answer is 11."
        },
        {
            "input": "3 3\n2 4 8\n6 14 16\n",
            "output": "6",
            "explanation": "Target 6 can be achieved with [2,4] (2
                crystals), target 14 can be achieved with [2,4,8] (3
                crystals), target 16 cannot be achieved. Minimum is 2
                crystals for target 6."
        }
    ],
    "tags_used": ["Subset Sum Problem", "Binary search", "bitmask", "
        array", "Heap"],
    "time_complexity": "O(2^n * k + k log k)",
    "space_complexity": "O(2^n + k)"
}
```

Figure 19: Problem 301's natural-language description and example unit test.

### F.1 Qualified Python Code Generated

```python
def magical_crystal_collection(n, k, crystals, targets):
    """
    Find the magical target value that can be achieved with minimum
        crystals.

    Args:
        n: number of crystals
        k: number of target values
        crystals: list of crystal power values
        targets: list of magical target values

    Returns:
        The target value achievable with minimum crystals, or -1 if none
            achievable
    """
    # Dictionary to store minimum crystals needed for each possible sum
    min_crystals_for_sum = {}

    # Generate all possible subsets using bit manipulation
    for mask in range(1, 1 << n):  # Start from 1 to exclude empty subset
        subset_sum = 0
        crystal_count = 0

        for i in range(n):
            if mask & (1 << i):
                subset_sum += crystals[i]
                crystal_count += 1

        # Update minimum crystals needed for this sum
        if subset_sum not in min_crystals_for_sum or crystal_count <
            min_crystals_for_sum[subset_sum]:
            min_crystals_for_sum[subset_sum] = crystal_count

    # Find the best target
    best_target = -1
    min_crystals_needed = float('inf')

    for target in targets:
        if target in min_crystals_for_sum:
            crystals_needed = min_crystals_for_sum[target]
            if (crystals_needed < min_crystals_needed or
                (crystals_needed == min_crystals_needed and target <
                    best_target)):
                min_crystals_needed = crystals_needed
                best_target = target

    return best_target
```

Figure 20: Problem 301 passes all 40 unit tests.

## F.2 Unqualified Python Code Generated

```python
def forest_fire_spread(n, firefighters, edges):
    """
    Find the maximum total number of firefighters that can be impacted
    when starting the fire from the optimal station.

    Args:
        n: number of stations
        firefighters: list of firefighters at each station (0-indexed)
        edges: list of tuples representing bidirectional trails

    Returns:
        Maximum total number of firefighters that can be impacted
    """
    # Build adjacency list
    graph = [[] for _ in range(n)]
    for u, v in edges:
        graph[u-1].append(v-1)  # Convert to 0-indexed
        graph[v-1].append(u-1)

    def dfs(start, visited, current_firefighters):
        """
        Perform DFS to spread fire from current station.
        Fire spreads to neighbor only if current station has more
            firefighters.
        """
        visited.add(start)
        total_impact = firefighters[start]

        for neighbor in graph[start]:
            if neighbor not in visited and current_firefighters > \
                firefighters[neighbor]:
                total_impact += dfs(neighbor, visited, firefighters[
                    neighbor])

        return total_impact

    max_impact = 0

    # Try starting fire from each station
    for start_station in range(n):
        visited = set()
        impact = dfs(start_station, visited, firefighters[start_station])
        max_impact = max(max_impact, impact)

    return max_impact
```

Figure 21: Problem 1547 has 22 passing and 19 failing test-cases. The DFS function incorrectly propagates state.

```python
def digital_library_traversal(input_str):
    """
    Performs preorder traversal of book dependencies starting from a
        given book.

    Args:
        input_str: String containing book dependencies and starting book

    Returns:
        String with books in preorder traversal order, one per line
    """
    lines = input_str.strip().split('\n')
    n = int(lines[0])

    # Parse dependencies
    dependencies = {}

    for i in range(1, n + 1):
        line = lines[i]
        if ':' in line:
            book, deps_str = line.split(':', 1)
            book = book.strip()
            deps_str = deps_str.strip()

            if deps_str:
                deps = [dep.strip() for dep in deps_str.split(',')]
                # Sort dependencies alphabetically
                deps.sort()
            else:
                deps = []

            dependencies[book] = deps

    starting_book = lines[n + 1].strip()

    # Check if starting book exists
    if starting_book not in dependencies:
        return "Book not found"

    # Perform preorder traversal
    result = []

    def preorder_traversal(book):
        # Visit current book
        result.append(book)

        # Visit dependencies in alphabetical order
        if book in dependencies:
            for dependency in dependencies[book]:
                if dependency in dependencies:
                    preorder_traversal(dependency)

    preorder_traversal(starting_book)

    return '\n'.join(result)
```

Figure 22: Problem 508 has 38 passing and 2 failing test-cases. Two unit tests violate the problem's stated constraints, preventing the otherwise-correct solution from passing all cases.

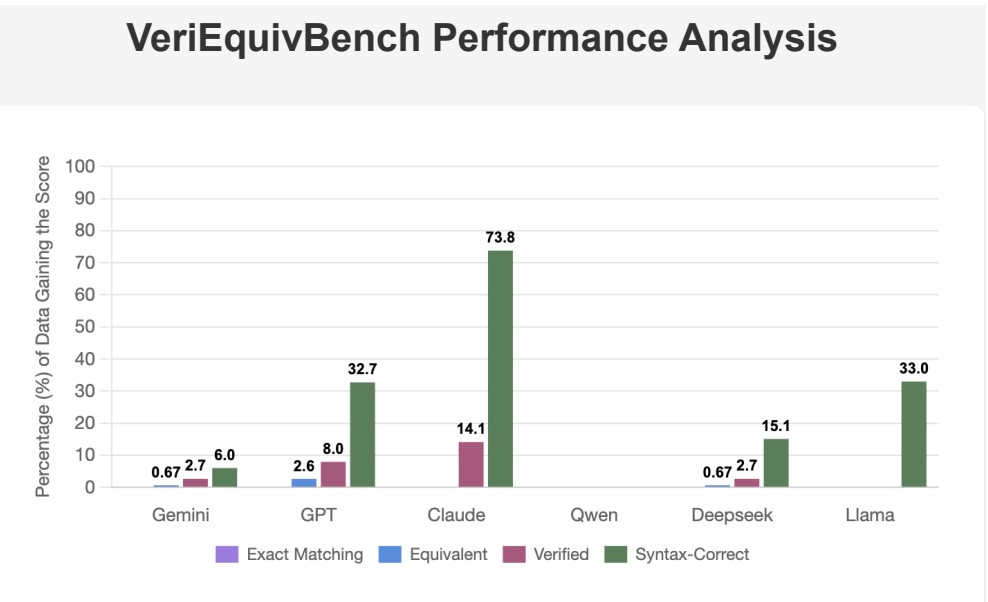

Figure 23: We evaluate three proprietary models and three open-sourced models using our pipeline to verify the alignment between users' intention and code generation. It turns out that business-purpose models in large sizes outperform small, open-sourced models, especially in writing less ambiguous specifications.

## G EXPERIMENTAL RESULT DETAILS

### G.1 QUALITY ASSURANCE OF PYTHON2DAFNY TRANSFORMATION

With $1,011$ questions randomly selected, we transform their unit tests in Python to Dafny. However, due to the mismatch in Dafny grammar, not all questions are successfully transformed without syntax errors; for example, our script uses **seq<int>** to define lists, but some Dafny code requires **array<int>**. Also, not all unit tests satisfy the preconditions, in which cases, the unverified Dafny code cannot be executed. Also, a few codes take too long to compile and are stopped after 30 minutes. The details are listed in Table 12.

Finally, with 648 successfully executed code, 530 pass all unit tests with a full pass rate at $81.79\%$.

Table 12: The table shows the success rate of transforming Python unit tests to Danfy.

| Selected Questions | Syntax Errors | Verification Errors | Timeout | Successful Execution |
|---|---|---|---|---|
| 1011 | 297 | 62 | 4 | 648 |
| 100% | 29.38% | 6.13% | 0.4% | 64.1% |

### G.2 TESTING ON MORE OPEN-SOURCED MODELS

We evaluate three proprietary models and three open-sourced models using our pipeline to verify the alignment between users' intentions and code generation. Here, we use Claude-4-sonnet, GPT-5, Gemini-2.5-flash, DeepSeek-R1, Qwen-2.5-Coder-14B-Instruct and Llama3-70B. Business-purpose models significantly outperform open-sourced models in writing syntax-correct and tight specifications.

### G.3 TRAINING CURVES ON AUXILIARY TASKS

We use the 14B SFT model provided by the Veri-Code Team and their code to RL-train models using GRPO.

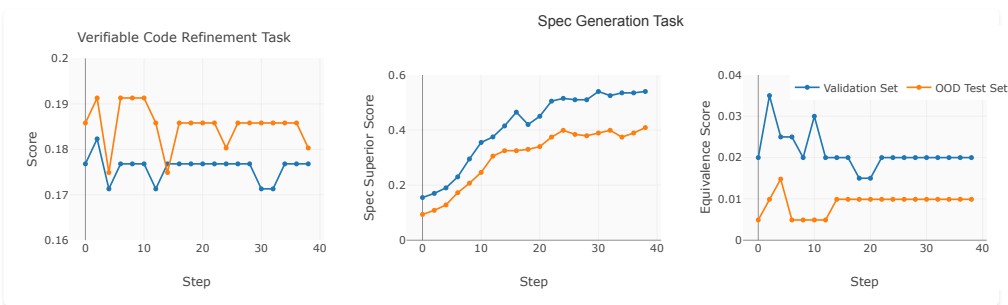

Figure 24: During the verifiable code refinement task, the model barely improves, demonstrating that RL training is not enough. During the spec generation task, the generated specification quality keeps enhancing, but still fails to capture code behaviours without ambiguities.

### G.4 Details about DafnySynthesis Inspection

This section details our analysis of 14 ground-truth samples identified as problematic. Our investigation revealed that two samples failed initial verification due to implementation errors or timeouts, precluding further analysis. These were #566 and #632, the latter of which was previously reported by Clover (Sun et al., 2024).

The primary issue in the remaining 12 samples was specification ambiguity stemming from insufficient post-conditions. We successfully rectified this in eight cases by strengthening their post-conditions, with the fixes validated through equivalence testing. Although our refinements improved the specifications for two other samples, they still did not pass the equivalence check. We were unable to resolve the ambiguities in the final two samples.

A significant portion of these ambiguous samples were newly discovered. Specifically, eight samples (#579, #602, #625, #629, #733, #755, #793, #807) were not documented in the prior work by Clover (Sun et al., 2024). Of these, we successfully fixed five (#625, #733, #755, #793, #807) and refined one (#602). Corresponding code examples are shown in Figures 25-32.

Regarding the issues previously reported by Clover, our findings for samples #567, #576, #644, and #803 largely concur. We fixed three (#567, #644, #803) and refined one (#576), with details in Figures 33-35. Conversely, sample #472 passed our equivalence checks, which aligns with Clover's classification of its issue as a natural-language inconsistency rather than a specification defect. As noted, sample #632 was excluded due to a timeout.

```
predicate InArray(a: array<int>, x: int)
    reads a
{
    exists i :: 0 <= i < a.Length && a[i] == x
}

method DissimilarElements(a: array<int>, b: array<int>) returns (result:
    seq<int>)
    ensures forall x :: x in result ==> (InArray(a, x) != InArray(b, x))
    ensures forall i, j :: 0 <= i < j < |result| ==> result[i] != result[
        j]
    ######## The post-conditions here omit that the first half of result
        is in array a, while the second half is in b.
{
    var res: seq<int> := [];
    for i := 0 to a.Length
        invariant 0 <= i <= a.Length
        invariant forall x :: x in res ==> InArray(a, x)
        invariant forall x :: x in res ==> InArray(a, x) != InArray(b, x)
        invariant forall i, j :: 0 <= i < j < |res| ==> res[i] != res[j]
    {
        if !InArray(b, a[i]) && a[i] !in res
        {
            res := res + [a[i]];
        }
    }

    ghost var partialSize := |res|;
    for i := 0 to b.Length
        invariant 0 <= i <= b.Length
        invariant forall k :: partialSize <= k < |res| ==> InArray(b, res
            [k])
        invariant forall k :: 0 <= k < |res| ==> InArray(a, res[k]) !=
            InArray(b, res[k])
        invariant forall i, j :: 0 <= i < j < |res| ==> res[i] != res[j]
    {
        if !InArray(a, b[i]) && b[i] !in res
        {
            res := res + [b[i]];
        }
    }

    result := res;
}
```

Figure 25: An unresolved specification ambiguity in DafnySynthesis sample #579. The post-condition is insufficient because it fails to enforce the preservation of the relative order of elements from the input array in the output.

```
method FindFirstRepeatedChar(s: string) returns (found: bool, c: char)
    ensures found ==> exists i, j :: 0 <= i < j < |s| && s[i] == s[j]
      && s[i] == c && (forall k, l :: 0 <= k < l < j
      && s[k] == s[l] ==> k >= i)
    ensures !found ==> (forall i, j :: 0 <= i < j < |s| ==> s[i] != s[j])
    ######## ⇓ The added post-condition
    ensures !found ==> c == ' '
    ######## ⇑
{
    c := ' ';
    found := false;
    var inner_found := false;
    var i := 0;
    while i < |s| && !found
        invariant 0 <= i <= |s|
        invariant found == inner_found

        invariant found ==> exists ii, jj :: 0 <= ii < i
          && ii < jj < |s| && s[ii] == s[jj] && s[ii] == c
          && (forall k, l :: 0 <= k < l < jj && s[k] == s[l] ==> k >= ii)

        invariant !found <==> (forall ii, jj :: 0 <= ii < i
          && ii < jj < |s| ==> s[ii] != s[jj])
        ######## ⇓ The added loop invariant
        invariant !found ==> c == ' '
        ######## ⇑
    {
        var j := i + 1;
        while j < |s| && !inner_found
            invariant i < j <= |s|
            invariant inner_found ==> exists k :: i < k < |s|
              && s[i] == s[k] && s[i] == c
            invariant !inner_found
              <==> (forall k :: i < k < j ==> s[i] != s[k])
            ######## ⇓ The added loop invariant
            invariant !inner_found ==> c == ' '
            invariant !found
            ######## ⇑
        {
            if s[i] == s[j] {
                inner_found := true;
                c := s[i];
            }
            j := j + 1;
        }
        found := inner_found;
        i := i + 1;
    }
}
```

Figure 26: A refined but unfixed specification for sample #602. While the shown refinement fails the equivalence test, a stricter post-condition (k > i) could not be verified due to a timeout.

```
method SwapFirstAndLast(a: array<int>)
    requires a.Length > 0
    modifies a
    ######## ⇓ The added post-condition
    ensures a.Length == old(a.Length)
    ######## ⇑
    ensures a[0] == old(a[a.Length - 1])
    ensures a[a.Length - 1] == old(a[0])
    ensures forall k :: 1 <= k < a.Length - 1 ==> a[k] == old(a[k])
{
    var tmp := a[0];
    a[0]  := a[a.Length - 1];
    a[a.Length - 1]  := tmp;
}
```

Figure 27: A successfully resolved specification ambiguity in DafnySynthesis sample #625. The original specification was ambiguous as it lacked a constraint on the output array's length. The ambiguity was rectified by introducing a post-condition ensuring the length remains invariant.

```
predicate IsEven(n: int)
{
    n % 2 == 0
}

method FindEvenNumbers(arr: array<int>) returns (evenList: seq<int>)

    ensures forall i :: 0 <= i < |evenList| ==> IsEven(evenList[i])
    && evenList[i] in arr[..]
    ensures forall i :: 0 <= i < arr.Length && IsEven(arr[i])
    ==> arr[i] in evenList
    ######## The post-conditions here do not ensures the order preserving
        between the input array and output array
{
    evenList := [];
    for i := 0 to arr.Length
        invariant 0 <= i <= arr.Length
        invariant 0 <= |evenList| <= i
        invariant forall k :: 0 <= k < |evenList| ==> IsEven(evenList[k])
            && evenList[k] in arr[..]
        invariant forall k :: 0 <= k < i && IsEven(arr[k]) ==> arr[k] in
            evenList
    {
        if IsEven(arr[i])
        {
            evenList := evenList + [arr[i]];
        }

    }
}
method FindEvenNumbers_check(arr: array<int>) returns (evenList: seq<int
    >)
{
  evenList := *;
  assume forall i :: 0 <= i < |evenList| ==> IsEven(evenList[i]) &&
      evenList[i] in arr[..];
  assume forall i :: 0 <= i < arr.Length && IsEven(arr[i]) ==> arr[i] in
      evenList;
  var val_0 :=FindEvenNumbers(arr);
  assert evenList[..] == val_0[..];

}
```

Figure 28: An unresolved specification ambiguity in DafnySynthesis sample #629. The post-condition is insufficient because it fails to enforce the preservation of the relative order of elements from the input array.

```
method FindFirstOccurrence(arr: array<int>, target: int) returns (index:
    int)
    requires arr != null
    requires forall i, j :: 0 <= i < j < arr.Length ==> arr[i] <= arr[j]
    ensures 0 <= index < arr.Length ==> arr[index] == target
    ensures index == -1 ==> forall i :: 0 <= i < arr.Length ==> arr[i] !=
        target
    ensures forall i :: 0 <= i < arr.Length ==> arr[i] == old(arr[i])
    ######## ⇓ The added post-condition
    ensures 0 <= index < arr.Length || index == -1
    ensures 0 <= index < arr.Length ==> ((forall i :: 0 <= i < index ==>
        arr[i] < arr[index]) && (forall j :: index <= j < arr.Length ==>
        arr[j] >= arr[index]))
    ######## ⇑
{
    index := -1;
    for i := 0 to arr.Length
        invariant 0 <= i <= arr.Length
        invariant index == -1 ==> forall k :: 0 <= k < i ==> arr[k] !=
            target
        invariant 0 <= index < i ==> arr[index] == target
        invariant forall k :: 0 <= k < arr.Length ==> arr[k] == old(arr[k
            ])
    {
        if arr[i] == target
        {
            index := i;
            break;
        }
        if arr[i] > target
        {
            break;
        }
    }
}
```

Figure 29: A successfully resolved specification ambiguity in DafnySynthesis sample #733. The original specification was insufficient, lacking detail for cases where the input index is non-negative. The issue was fixed by refining the post-condition to explicitly define the expected behavior for this scenario.

```
function MinPair(s: seq<int>) : (r: int)
    requires |s| == 2
    ensures s[0] <= s[1] <==> r == s[0]
    ensures s[0] > s[1] ==> r == s[1]
{
    if s[0] <= s[1] then s[0] else s[1]
}
function min(s: seq<int>) : (r: int)
    requires |s| >= 2
    ensures forall i :: 0 <= i < |s| ==> r <= s[i]
{
    if |s| == 2 then MinPair(s)
    else MinPair([s[0], min(s[1..])])
}
method SecondSmallest(s: array<int>) returns (secondSmallest: int)
    requires s.Length >= 2
    requires exists i, j :: 0 <= i < s.Length && 0 <= j < s.Length
      && i != j && s[i] == min(s[..]) && s[j] != s[i]
    ensures exists i, j :: 0 <= i < s.Length && 0 <= j < s.Length
      && i != j && s[i] == min(s[..]) && s[j] == secondSmallest
    ensures forall k ::  0 <= k < s.Length && s[k] != min(s[..])
      ==>  s[k] >= secondSmallest
    ######## ⇓ The added post-condition
    ensures (exists i, j :: i != j && 0 <= i < s.Length
      && 0 <= j < s.Length && s[i] == s[j] && s[i] == min(s[..]))
      ==> secondSmallest == min(s[..])
    ensures !(exists i, j :: i != j && 0 <= i < s.Length
      && 0 <= j < s.Length && s[i] == s[j] && s[i] == min(s[..]))
      ==> ( (exists k :: 0 <= k < s.Length && s[k] == secondSmallest)
        && (forall k :: 0 <= k < s.Length && s[k] > min(s[..])
        ==> s[k] >= secondSmallest) && secondSmallest > min(s[..]) )
    ######## ⇑
{
    var minIndex := 0;
    var secondMinIndex := 1;
    if s[1] < s[0] {
        minIndex := 1;
        secondMinIndex := 0;
    }
    for i := 2 to s.Length
    invariant 0 <= i <= s.Length
    invariant 0 <= minIndex < i
    invariant 0 <= secondMinIndex < i
    invariant minIndex != secondMinIndex
    invariant forall k :: 0 <= k < i ==> s[k] >= s[minIndex]
    invariant forall k :: 0 <= k < i && k != minIndex ==> s[k] >= s[
        secondMinIndex]
    {
        if s[i] < s[minIndex] {
            secondMinIndex := minIndex;
            minIndex := i;
        } else if s[i] < s[secondMinIndex] {
            secondMinIndex := i;
        }
    }

    secondSmallest := s[secondMinIndex];
}
```

Figure 30: A successfully resolved specification ambiguity in DafnySynthesis sample #755. The original specification was insufficient, failing to distinguish between cases with a unique minimum value and those with multiple occurrences of the minimum. The ambiguity was rectified by refining the post-condition to explicitly detail the expected behavior for both scenarios.

```
method LastPosition(arr: array<int>, elem: int) returns (pos: int)
    requires arr.Length > 0
    requires forall i, j :: 0 <= i < j < arr.Length ==> arr[i] <= arr[j]
    ######## ⇓ Original post-condition
    // ensures pos == -1 || (0 <= pos < arr.Length && arr[pos] == elem &&
        (pos <= arr.Length - 1 || arr[pos + 1] > elem))
    ######## ⇑
    ######## ⇓ The fixed post-condition
    ensures pos == -1 <==> (forall j :: 0 <= j < arr.Length ==> arr[j] !=
        elem)
    ensures pos != -1 <==> (0 <= pos < arr.Length && arr[pos] == elem &&
        (pos == arr.Length - 1 || arr[pos + 1] > elem))
    ######## ⇑
    ensures forall i :: 0 <= i < arr.Length ==> arr[i] == old(arr[i])
{
    pos := -1;
    for i := 0 to arr.Length  #### Originally, the upper bound is arr.
        Length - 1, but it was buggy
        invariant 0 <= i <= arr.Length
        ######## ⇓ Original loop invariant
        // invariant pos == -1 || (0 <= pos < i && arr[pos] == elem && (
            pos == i - 1 || arr[pos + 1] > elem))
        ######## ⇓ The fixed loop invariant
        invariant pos == -1 <==> (forall j :: 0 <= j < i ==> arr[j] !=
            elem)
        invariant pos != -1 <==> (0 <= pos < i && arr[pos] == elem && (
            pos == i - 1 || arr[pos + 1] > elem))
        ######## ⇑
        invariant forall k :: 0 <= k < arr.Length ==> arr[k] == old(arr[k
            ])
    {
        if arr[i] == elem
        {
            pos := i;
        }
    }
}
```

Figure 31: A successfully resolved specification ambiguity in DafnySynthesis sample #793. The original specification was insufficient as it failed to define distinct behaviors based on the sign of the input parameter 'pos'. The ambiguity was rectified by refining the post-condition to explicitly handle the cases where 'pos' is negative and non-negative, respectively.

```
predicate IsOdd(x: int)
{
    x % 2 != 0
}

method FindFirstOdd(a: array<int>) returns (found: bool, index: int)
    requires a != null
    ensures !found ==> forall i :: 0 <= i < a.Length ==> !IsOdd(a[i])
    ensures found ==> 0 <= index < a.Length && IsOdd(a[index])
        && forall i :: 0 <= i < index ==> !IsOdd(a[i])
    ######## ⇓ The added post-condition
    ensures !found ==> index == a.Length
    ######## ⇑
{
    found := false;
    index := 0;
    while (index < a.Length)
        invariant 0 <= index <= a.Length
        invariant !found ==> forall i :: 0 <= i < index ==> !IsOdd(a[i])
        invariant found ==> IsOdd(a[index - 1]) && forall i :: 0 <= i <
            index - 1 ==> !IsOdd(a[i])
    {
        if IsOdd(a[index])
        {
            found := true;
            return;
        }
        index := index + 1;
    }
}
```

Figure 32: A successfully resolved specification ambiguity in DafnySynthesis sample #807. The original specification was insufficient, as it only described the behavior for successful outcomes. The ambiguity was resolved by strengthening the post-condition to explicitly define the program's state in failure cases, ensuring comprehensive and predictable behavior.

```
method IsSorted(a: array<int>) returns (sorted: bool)
    requires a.Length > 0
    ######## ⇓ Original post-condition
    // ensures sorted <== forall i, j :: 0 <= i < j < a.Length
      ==> a[i] <= a[j]
    // ensures !sorted ==> exists i, j :: 0 <= i < j < a.Length
      && a[i] > a[j]
    ######## ⇑
    ######## ⇓ The fixed post-condition
    ensures sorted <==> forall i, j :: 0 <= i < j < a.Length
      ==> a[i] <= a[j]
    ######## ⇑
{
    sorted := true;
    for i := 0 to a.Length - 1
        invariant 0 <= i < a.Length
        ######## ⇓ Original loop invariant
        // invariant sorted <== forall k, l :: 0 <= k < l < i
          ==> a[k] <= a[l]
        // invariant !sorted ==> exists k :: 0 <= k < i && a[k] > a[k+1]
        ######## ⇑
        ######## ⇓ The fixed post-condition
        invariant sorted <==> forall k, l :: 0 <= k < l <= i
          ==> a[k] <= a[l]
        ######## ⇑

    {
        if a[i] > a[i + 1]
        {
            sorted := false;
            break;
        }
    }
    sorted := sorted;
}
```

Figure 33: A successfully resolved specification ambiguity in DafnySynthesis sample #567, an issue also identified by the Clover. The original post-condition was overly permissive, stating only a sufficient condition for the desired outcome. The ambiguity was rectified by strengthening this to a necessary and sufficient condition (an equivalence).

```
method Reverse(a: array<int>)
  modifies a
    ######## ⇓ The added post-condition
  ensures a.Length == old(a.Length)
    ######## ⇑
  ensures forall k :: 0 <= k < a.Length ==> a[k] == old(a[(a.Length-1) -
      k])
{
  var l := a.Length - 1;
  var i := 0;
  while (i < l-i)
    invariant 0 <= i <= (l+1)/2
    invariant forall k :: 0 <= k < i || l-i < k <= l ==> a[k] == old(a[l-
        k])
    invariant forall k :: i <= k <= l-i ==> a[k] == old(a[k])
  {
    a[i], a[l-i] := a[l-i], a[i];
    i := i + 1;
  }
}
method ReverseUptoK(s: array<int>, k: int)
    modifies s
    requires 2 <= k <= s.Length
    ######## ⇓ The added post-condition
    ensures s.Length == old(s.Length)
    ######## ⇑
    ensures forall i :: 0 <= i < k ==> s[i] == old(s[k - 1 - i])
    ensures forall i :: k <= i < s.Length ==> s[i] == old(s[i])
{
  var l := k - 1;
  var i := 0;
  while (i < l-i)
    invariant 0 <= i <= (l+1)/2;
    invariant forall p :: 0 <= p < i || l-i < p <= l ==> s[p] == old(s[l-
        p]);
    invariant forall p :: i <= p <= l-i ==> s[p] == old(s[p]);
        invariant forall p :: k <= p < s.Length ==> s[p] == old(s[p])
  {
    s[i], s[l-i] := s[l-i], s[i];
    i := i + 1;
  }
}
```

Figure 34: A successfully resolved specification ambiguity in DafnySynthesis sample #644, an issue also identified by the Clover. The original specification was ambiguous as it lacked a constraint on the output array's length. The ambiguity was rectified by introducing a post-condition ensuring the length remains invariant.

```
method IsPerfectSquare(n: int) returns (result: bool)
    requires n >= 0
    ######## ⇓ Original post-condition
    // ensures result == true ==> (exists i: int :: 0 <= i <= n && i * i
        == n)
    // ensures result == false ==> (forall a: int :: 0 < a*a < n ==> a*a
        != n)
    ######## ⇑
    ######## ⇓ The fixed post-condition
    ensures result <==> (exists i: int :: 0 <= i <= n && i * i == n)
    ######## ⇑
{
    var i := 0;
    while (i * i < n)
        invariant 0 <= i <= n
        invariant forall k :: 0 <= k < i ==> k * k < n
    {
        i := i + 1;
    }
    return i * i == n;
}
```

Figure 35: A successfully resolved specification ambiguity in DafnySynthesis sample #803, an issue also identified by the Clover. The original post-condition was overly permissive, stating only necessary conditions for the desired outcome. The ambiguity was rectified by strengthening this to a necessary and sufficient condition (an equivalence).

```
method IsSublist(sub: seq<int>, main: seq<int>) returns (result: bool)
    ######## ⇓ Original post-condition
    // ensures true <== (exists i :: 0 <= i <= |main| - |sub| && sub ==
        main[i..i + |sub|])
    ######## ⇑
    ######## ⇓ The refined post-condition
    ensures result ==> (exists i :: 0 <= i <= |main| - |sub| && sub ==
        main[i..i + |sub|])
    ensures result ==> (exists i :: |sub| <= i <= |main| && sub == main[i
        - |sub|..i])
    ######## ⇑
{
    if |sub| > |main| {
        return false;
    }
    result := false;
    for i := 0 to |main| - |sub| + 1
        ######## ⇓ The original loop invariant
        // invariant result ==> (exists j :: 0 <= j < i && sub == main[j
            ..j + |sub|])
        ######## ⇑
        ######## ⇓ The refined loop invariant
        invariant 0 <= i <= |main| - |sub| + 1
        ######## ⇑

    {
        if sub == main[i..i + |sub|] {
            result := true;
        }
    }
    result := false;
}
```

Figure 36: An unresolved specification ambiguity in DafnySynthesis sample #576, an issue also identified by the Clover. The original post-condition was effectively meaningless, providing no meaningful constraints. Although the post-condition was refined to be more specific, the resulting specification still fails to pass the equivalence test, indicating that the ambiguity has not been fully resolved and requires further investigation.

## H    The Use of Large Language Models

Multiple LLM products, including GPT-5 and Gemini-2.5-pro, are deployed to polish the writing. However, none of the paragraphs is written by LLMs directly, and all research ideas are independently proposed by authors without any AI assistance. Claude-Opus-4.1 and Sonnet are used to create figure generation code for Figure 5 and 24. Cursor is included to assist coding, but all generated code is then carefully inspected by authors. Other uses of LLMs in data curation and synthesis are clearly stated in the paper.

