# OpenReview forum: "VeriEquivBench: An Equivalence Score for Ground-Truth-Free Evaluation of Formally Verifiable Code"
_ICLR.cc/2026/Conference — ICLR 2026 Poster_

### Official Review · Reviewer_LnLN · 2025-10-31

**Soundness:** 2
**Presentation:** 2
**Contribution:** 3
**Rating:** 4
**Confidence:** 3

**Summary:**

This paper addresses the formal verification of LLM-generated code, specifically within the Dafny language. The authors introduce two primary contributions: VeriEquivBench, a new large-scale benchmark of 2,389 algorithmic problems, which is significantly larger than prior datasets. The Equivalence Score, a novel, ground-truth-free metric. This metric uses the Dafny verifier to formally check for bidirectional implication between generated code and its specifications. The paper concludes with an evaluation of state-of-the-art LLMs on this new benchmark, highlighting the significant difficulty of the task.

**Strengths:**

The proposed Equivalence Score is a significant conceptual contribution. By formally checking for bidirectional implication, it removes the dependency on manually-authored ground-truth specifications. This provides a more rigorous and automated method for evaluating the quality and completeness of the generated specifications themselves, which is a key bottleneck in this area.

The scale and complexity of VeriEquivBench are commendable. The dataset's 2,389 problems represent a more than 10x increase over existing benchmarks like DafnySynthesis and CloverBench combined. The higher average Cyclomatic Complexity (5.63 vs. 2.44) confirms that the problems are more challenging.

The inclusion of the TagComp subset, generated via a tag-composition pipeline, is a good step towards creating novel, contamination-free problems for evaluation.

**Weaknesses:**

Misleading Explanation of the Equivalence Score: The paper's explanation of the equivalence score is inaccurate. The authors claim the metric verifies if the method's output is the "unique value" satisfying the specifications. A verifier like Dafny proves correctness (i.e., that the code's output is a valid value that satisfies the post-conditions), not uniqueness. While the goal is clearly to ensure the specification is "tight" (i.e., not under-specified), describing this as a check for a "unique value" misrepresents the verifier's function and the metric's actual technical contribution.

Flawed Natural Language Alignment Check: The paper's second evaluation step, which translates formal specifications back to natural language (FL-to-NL) and uses an LLM-as-a-judge, has several issues. First, it re-introduces a non-formal, unreliable evaluation method, which undermines the paper's primary goal of achieving formal rigor. The reliability of this LLM judge is not validated against human assessments. Second, this FL-to-NL evaluation approach is not novel and is a known technique in related autoformalization communities (e.g., for Lean), which the paper fails to discuss [1].

Insufficient Validation of Benchmark Data: The quality and correctness of the VeriEquivBench dataset are not adequately established. The authors report that the Python-to-Dafny code transformation was validated by running translated unit tests on only 20 random samples (out of 2,389). While 90% of these passed, this sample size (<1%) is far too small to provide statistical confidence in the correctness of the entire 2,389-problem benchmark. Citing "long compilation times" is not a sufficient justification for such a limited validation.

[1] Ying H, Wu Z, Geng Y, et al. Lean workbook: A large-scale lean problem set formalized from natural language math problems[J]. Advances in Neural Information Processing Systems, 2024, 37: 105848-105863.

**Questions:**

The paper presents the Equivalence Score as a rigorous check for code-specification alignment. If a pair passes this bidirectional implication check, the specification should, by definition, precisely and completely capture the behavior of the code. Why, then, is the secondary, unreliable Spec -> NL -> LLM-Judge step necessary? Does its inclusion imply that the Equivalence Score alone is insufficient to guarantee that the specification aligns with the original natural language query intent?

The TagComp pipeline is presented as a "scalable method" for generating novel problems. However, the paper's own data shows a very low yield: from ~1,900 initial problem descriptions, only 300 (~15.8%) had Python solutions that passed the unit tests, and only 215 (~11.3%) cleared the full pipeline. Given this low success rate, in what sense is this pipeline considered "scalable"? Furthermore, what analysis was done to determine the primary failure mode? Why did ~84% of the LLM-synthesized Python solutions fail to pass their own corresponding unit tests?

---

> ### Author Response · Authors · 2025-11-19
> **Clarifications on Reviewer’s Questions**
>
> **Dear Reviewer LnLN**
>
> We appreciate your detailed review and helpful suggestions. Thank you for recognizing our equivalence score as a significant contribution and acknowledging the size and complexity of our formal verification benchmark. Below, we address your questions and concerns point by point.
>
> 1. Misleading Explanation of the Equivalence Score:
>
>     Thank you for pointing out the inaccurate description of the equivalence score. We have corrected all misleading explanations on lines 297 and 308. As the reviewer notes, the contribution itself remains rigorous, supported by thorough analysis.

---

> ### Author Response · Authors · 2025-11-19
> **Clarification on Flawed Natural Language Alignment Check**
>
> 2. Flawed Natural Language Alignment Check and Q1:
>
> **(1) Spec -> NL -> LLM-Judge is non-formal and unreliable:**
>
> Since coding problems are described in natural language, this part of the pipeline inherently involves non-formal reasoning. However, as shown by the three examples in Appendices E.2, E.3, and E.4, current proprietary LLMs are capable of translating specifications back into natural language and identifying subtle differences. For instance, in E.3, we construct a hard example where only one post-condition on connectivity is removed, and the pipeline correctly detects the change. In E.4, the generated code is incorrect even though the specifications are tight, and the pipeline also identifies this. The reliability of this step comes from the relative simplicity of translating spec to NL, our prompt engineering, and the capabilities of current LLMs.
>
> **(2) Novelty of Spec-to-NL translation:**
>
> We have updated the paper to correctly credit prior work (line 092). While the Spec-to-NL translation itself is not novel, our overall pipeline for verifying alignment with the user’s intention is both novel and important. To our knowledge, this is the first pipeline that checks alignment without any human annotation.
>
> **(3) Why is the Spec -> NL -> LLM-Judge step necessary?**
>
> Although the equivalence score establishes alignment between the code and the formal specification, it does not guarantee equivalence to the natural language problem description. As shown in Appendix E.4, the code can be incorrect even when the specification accurately describes it. Thus, this additional step is needed to close the gap between the formal specification and the original NL query.

---

> ### Author Response · Authors · 2025-11-19
> **Other Questions**
>
> 3. Insufficient Validation of Benchmark Data:
>
>     We developed a script to transform LeetCode Python unit tests into Dafny tests. Among the 648 transformed problems, 81.79% pass all official LeetCode unit tests. Details of the transformation are provided in Appendix G.1. This demonstrates the high quality of our Python-to-Dafny conversion.
>
> 4. Scalability of the TagComp pipeline:
>
>     (1) Our pipeline synthesizes 1,893 novel algorithmic problems with slightly greater complexity than LeetCode, as shown in Table 3. With our auto-evaluation pipeline, no Python code, unit tests, or Dafny specifications are required for new tasks. This **tag composition** and **auto-evaluation** framework shows strong potential for scalability.
>
>     (2) Due to the novelty of our generated problems, not all can be solved by current LLMs. As shown in Appendix F, errors in either the generated code or unit tests cause many LLM-produced answers to be invalid. Consequently, 85 out of 300 samples fail to produce either (1) equivalent but verifiable specifications for the NL description or (2) verifiable code when the specifications are overly loose.

---

> ### Author Response · Authors · 2025-11-25
>
> Dear Reviewer,
>     We hope this message finds you well. As the discussion period is nearing its end, we would like to confirm that we have satisfactorily addressed all your concerns. If any additional points or feedback remain, please let us know. Your insights are invaluable, and we are eager to resolve any outstanding issues to improve our work.
>     Thank you very much for your time and effort in reviewing our paper.

---

### Official Review · Reviewer_Bjj7 · 2025-11-03

**Soundness:** 1
**Presentation:** 1
**Contribution:** 2
**Rating:** 2
**Confidence:** 4

**Summary:**

The paper proposes an alternative setting of verifiable code generation (jointly generating code, formal specifications, and proofs). To evaluate LLM-generated specs, prior work often checks their equivalence with human-annotated ground-truth specs, which are expensive to collect. This work tries to avoid that by using Dafny to prove the logical equivalence of the generated code and spec. The spec is considered correct if and only if it is equivalent to the code.

**Strengths:**

* The authors collected a new benchmark for verifiable code generation, which can be useful.

**Weaknesses:**

The paper's central contribution is a setup to evaluate specs by checking their logical equivalence with code, which is fundamentally flawed. A good spec often captures only a subset (rather than all) of the code's properties. Consider the task of writing a sorting algorithm, an LLM can easily generate a spec (the output list should be sorted, and it should contain exactly the same set of elements as the input list) and the code of a merge sort algorithm. However, under this paper's setup, this is incorrect because the spec fails to capture all properties of the code (e.g., merge sort is a stable sorting algorithm). This is not a contrived special case. In general, specs are at a higher level of abstraction than code and are expected to capture properties that are relevant.

I'm open to the authors' clarification if I had misunderstood anything.

**Questions:**

N/A

---

> ### Author Response · Authors · 2025-11-13
> **Motivation Clarification**
>
> Dear Reviewer Bjj7
>
> Thanks for your question!
>
> Your example turns out to support the necessity of an equivalence score. Given the post-conditions of a sorted list with exactly the same set of elements, the code can double the size of the original input. For example, for [3, 1, 2], the code returns [1, 1, 2, 2, 3, 3]. It is difficult to determine whether the specs are good or not if they are not tight.
>
> Our benchmark targets a fundamental key problem: how to automatically verify that the code is correct and aligned with the user's intention. That is why we need specifications that unambiguously describe what the code does, avoiding any edge cases, as you provide. Before our benchmark, there was no established way to verify alignment with the user's intention in terms of scalability, because both "subset" formal specifications and unit tests require careful design, which necessitates significant effort, but cannot guarantee the checking of edge cases.
>
> Meanwhile, we would like to thank you for acknowledging our collection of a new verifiable code benchmark. As other reviewers have pointed out, our new collection is not only significantly larger but also more complex compared to algorithms like merge sort, probing the boundaries of current proprietary LLMs.
>
> Please let us know if we have addressed your primary concern regarding motivation. If you have any other concerns or suggestions, we sincerely appreciate your time and assistance in helping us improve the paper. We would be more than happy to discuss them further.

---

> ### Comment · Reviewer_Bjj7 · 2025-11-20
>
> To clarify: by `it should contain exactly the same set of elements as the input list`, I actually meant the same "multiset".
>
> The real question is: The user says "I want a sorting algorithm", and we write down the formal spec of a sorting algorithm as `s`. Because the user didn't specify whether they want a stable sorting algorithm or not, `s` shouldn't capture any property about stability. Then, the model generates code `c` to implement the spec. `c` is either stable or unstable. If `c` is stable, it is wrong according to the definition in this paper, because `c` has additional properties (stability) not in `s`. Similarly, if `c` is unstable, it is also wrong, because it has additional properties (instability) not in `s`.
>
> Conclusion: Under the definition of correctness in this paper, it is impossible for the model to generate a correct implementation  when user simply wants a sorting algorithm.
>
> Any clarification from the authors is appreciated.

---

> > ### Author Response · Authors · 2025-11-21
> > **Serious Misunderstanding**
> >
> > Thank you for your feedback and the discussion! We believe there is a serious misunderstanding in your interpretation of our correctness definition, and we would like to ask a few clarification questions to ensure we're aligned.
> >
> > **“If `c` is stable, it is wrong according to the definition in this paper.”**
> >
> > We respectfully disagree: the paper does not provide a definition of code correctness based on specifications. We do not claim the code is wrong in your example, whether the user specifies the stability condition or not.
> >
> > Given a user query, our system can generate any code implementation; however, the **equivalence score** requires generating **tight and unambiguous** specifications simultaneously to precisely describe what the code does. Code paired with ambiguous specifications will be rejected, not because the code is “wrong”, but because it cannot be **formally verified**.
> >
> > Next, our pipeline translates these unambiguous specifications back into natural language. Both stable and unstable versions can pass the equivalence score if the specification describes them correctly.
> >
> > Finally, to make our benchmark auto-evaluable at scale, we employ an LLM-as-a-Judge, avoiding any reliance on human annotation. As demonstrated in several examples in **Appendix E**, subtle semantic differences are detectable during the spec-to-NL translation, and the LLM judgment can catch incorrect implementations.
> >
> > **“A good spec often captures only a subset (rather than all) of the code's properties.”**
> >
> > Previously, you defined “subset” specs as ones not capturing *all* properties of the code.
> >
> > We’d be grateful if you could elaborate: under what formal or empirical foundation do you define “good spec” in this way? In particular, how do you see ambiguity in the user’s query influencing the “goodness” of a spec in formal-verification settings?
> >
> > Building unambiguous specifications has long been a goal in the formal verification community, as seen in benchmarks such as **CLEVER** (Dafny) and **VERINA** (Lean). However, previous benchmarks rely heavily on human annotations, which limits their scalability. That is precisely why reviewers **dTTt** and **LnLN** consider our contribution to be clever, novel, and significant.
> >
> > [1] Sun, C., Sheng, Y., Padon, O., & Barrett, C. (2024, July). Clover: Closed-loop verifiable code generation. In *International Symposium on AI Verification* (pp. 134-155). Cham: Springer Nature Switzerland.
> >
> > [2] Ye, Z., Yan, Z., He, J., Kasriel, T., Yang, K., & Song, D. (2025). VERINA: Benchmarking Verifiable Code Generation. *arXiv preprint arXiv:2505.23135*.

---

> ### Comment · Reviewer_Bjj7 · 2025-11-25
>
> > Code paired with ambiguous specifications will be rejected, not because the code is “wrong”, but because it cannot be formally verified.
>
> I think we're aligned on this point. Just to clarify, "wrong" in my previous comment actually meant "unable to be verified". So I was thinking: "Under the setting in this paper, if the user simply wants a sorting algorithm and does not care if the algorithm is stable or not, it's impossible for the model to generate a formally verifiable solution."
>
>
> > under what formal or empirical foundation do you define “good spec” in this way? In particular, how do you see ambiguity in the user’s query influencing the “goodness” of a spec in formal-verification settings?
>
> What I said was that a good spec can be incomplete. In other words, the user may care about only certain properties while ignoring others. One example would be stable vs. unstable sorting algorithms we just discussed. I'm not sure if this is what "ambiguity" means in the authors' terminology.
>
> > Building unambiguous specifications has long been a goal in the formal verification community
>
> From my understanding, what the formal verification community wants is: ideally, the spec should fully capture the user's intent (, which is different from capturing the full properties of the code). If the spec is always required to be completely equivalent to the code, writing specs will be as hard as writing code.

---

> ### Author Response · Authors · 2025-11-26
>
> Please allow us to clarify this further.
>
> Our equivalence score is defined **with respect to the provided code**.
>
> If the code is stable, the specification should state its stability; if it is unstable, the specification should not claim stability. It is not related to the user's query at this stage.
>
> Both types of code with equivalent specifications are accepted, giving a high match with the user’s intention. Even better, in the translation of specifications, this feature, which is not mentioned in the user query, can be brought to the user’s attention.
>
> As long as there is a code implementation, there is a corresponding set of equivalent specifications that contains the strongest post-conditions.
>
> The equivalent specifications are required in our pipeline to automatically evaluate the quality of agent-generated specifications and prepare the code for intention alignment verification. Otherwise, the quality of generated specifications cannot be checked, and the code behaviour is not unambiguously stated.

---

### Official Review · Reviewer_dTTt · 2025-11-04

**Soundness:** 3
**Presentation:** 4
**Contribution:** 3
**Rating:** 6
**Confidence:** 4

**Summary:**

This paper introduces VeriEquivBench, a benchmark of 2K+ Dafny problems for evaluating formally verifiable code generation. The key innovation is an equivalence score that replaces ground-truth specification matching with bidirectional formal verification. Assuming the bidirectional formal verification is successful, this process eliminates the need for expensive manual specification annotation, which has limited prior benchmarks to much fewer problems.  The majority of the benchmark is from LeetCode, and a about 8% are synthetically generated. Empirical evaluations on three closed models show that there is a huge gap in performance.

**Strengths:**

The use of the equivalence score is clever, and as far as I know, quite novel.  The method is able to discover errors in previously manually specified ground true specs. The scale of the benchmark dwarfs previous benchmarks in this space, owing to its automated nature.

**Weaknesses:**

The paper's most significant limitation is its fairly thin empirical evaluation. Only three proprietary models are tested on the main task, with no evaluation of open-source models like DeepSeek-Coder, Qwen, or CodeLlama.  The paper also doesn't do much analysis of the failure modes of the models, beyond report a very low success rate.

Finally, I wonder about the diversity of leetcode questions.  My understanding is that the focus is on very algorithmic coding problems.  So while the benchmark is large in size, it may not be as diverse as one would ideally want.  (To be clear, I don't think this is a serious weakness, but worth commenting on.)

**Questions:**

How were the coding agents set up and prompted?

Can you comment on the types of coding problems that appear in LeetCode?

---

> ### Author Response · Authors · 2025-11-19
> **Experiments and Analysis**
>
> **Dear Reviewer dTTt,**
>
> We are glad to hear your positive comments on the significance and novelty of our equivalence score! We would like to further emphasize its importance in establishing the first automated evaluation pipeline for checking alignment with the query. Without relying on human-designed unit tests or annotations, our evaluation pipeline—together with the problem synthesis pipeline—offers the potential for large-scale application. We also appreciate your constructive suggestions regarding additional details that could strengthen the submission, and we have revised the paper accordingly.
>
> **Evaluation of Open-Source Models**
>
> We evaluated business-purpose models first, anticipating stronger performance. Following your suggestion, we have now included results for three open-source models across different scales: DeepSeek-R1-671B, CodeLlama-70B, and Qwen-2.5-Coder-14B-Instruct. Their performance, as shown in Figure 23 in Appendix G.2, remains consistently below the closed-source models, confirming the challenging nature of the task.
>
> **Failure Mode Analysis**
>
> Regarding the failure modes, the main challenge lies in generating **verifiable specifications**. As the examples in Appendix E.1 illustrate, verification often fails due to missing intermediate clauses. We also observe limitations from the Dafny verifier itself, which sometimes requires specifications to be written in particular forms; we provide two examples to demonstrate this.
>
> In addition, Appendices E.2–E.4 present cases with ambiguous specifications or incorrect code paired with tight specifications that nevertheless fail to align with the user’s intended natural-language description. Importantly, all of these cases are detected by our spec-to-NL and LLM-as-a-Judge pipeline, which ensures equivalence between the specification and the natural-language query.
>
> Appendix F further presents failure examples from the synthesized-data pipeline. Although we have generated 1,893 novel problems, 84.15% of the data contain either incorrect code or incorrect unit tests, highlighting the difficulty of code generation on our benchmark. This demonstrates that the benchmark challenges not only the writing of formal specifications but also the generation of correct executable code.

---

> > ### Author Response · Authors · 2025-11-19
> > **Clarifications on Reviewer’s Questions**
> >
> > **Q1: Model Setup**
> >
> > The coding agent utilizes in-context learning for improved code generation and multi-turn dialogue for error handling, without external tools. We configured temperature/top-p at 0.7/0.9 for spec generation and 0.5/0.8 for code synthesis and model evaluation, balancing creativity and stability. Detailed prompts and model arguments are in Appendices C and D.
> >
> > **Q2: Diversity of LeetCode Problems**
> >
> > We intentionally use algorithmic puzzles as a controlled testbed to evaluate models’ core reasoning capabilities rather than their engineering abilities. Meanwhile, LeetCode problems provide high-quality code, unit tests, and algorithm labels, supported by an active community, which substantially reduces the cost and complexity of formal annotation and makes such a comprehensive benchmark feasible. Although our focus is on algorithms, our dataset (including synthesized problems) spans nearly all major algorithmic categories and the full LeetCode difficulty spectrum, as detailed in Appendix A.

---

> ### Author Response · Authors · 2025-11-25
>
> Dear Reviewer,
>     We hope this message finds you well. As the discussion period is nearing its end, we would like to confirm that we have satisfactorily addressed all your concerns. If any additional points or feedback remain, please let us know. Your insights are invaluable, and we are eager to resolve any outstanding issues to improve our work.
>     Thank you very much for your time and effort in reviewing our paper.

---

### Author Response · Authors · 2025-12-04
**Summary for AC**

**Dear Area Chair,**

We sincerely thank you for taking the time to review our paper in this situation.

We also appreciate and respect previous reviewers’ engagement and constructive suggestions.

We would like to highlight the core contributions of our work and summarize the major revisions we have made:

**Core Contributions**

1. Our paper introduces an equivalence score to automatically verify the quality of generated formal specifications without human effort. This technical innovation is considered smart, clever and significant by **Reviewer dTTt and Reviewer LnLN.**
2. Based on the automatic evaluation of generated specifications, we propose a verifiable coding-agent flow to verify alignment with users’ intentions.
3. Thanks to the equivalence score, we provide a benchmark 10 times larger than previous benchmarks, with a substantial increase in code complexity, to evaluate whether agents can pass our test-time verification flow.
4. We also design another pipeline to synthesize novel problems by composition. With our automatic test-time verification flow, our benchmark allows scalability in both data size and complexity.

**Rebuttal Responses**

In response to **Reviewer dTTt**, which called for additional experiments on open-sourced models and analysis of failure modes, we have conducted the requested experiments and added **15 pages** of detailed results and failure case analyses. These demonstrate the reliability of both our verifiable coding pipeline and our data synthesis method.

**Reviewer LnLN** pointed out two detailed points for clarification: a more accurate description of our contributions and better attribution to related work. We have addressed both. Additionally, **Reviewer LnLN** raised concerns about the reliability of verifying user intent by translating formal specifications back to natural language and the quality of our Python-to-Dafny data transformation. To address this, we provided a thorough **qualitative analysis** of our test-time verification flow and a **quantitative study** validating transformation quality.

**Concerns**

We would also like to respectfully raise a concern regarding  **Reviewer Bjj7**’s comments.

We respectfully believe that the rejection rests on **a personal belief about what a method “should” or “should not” include**, rather than on technical or empirical evidence. Several critiques rely on undefined or vague terminology without contextual grounding, and some comments invoke concepts not used in the paper (e.g., “correctness of code”), suggesting a potential misunderstanding of our contribution.

The reviewer argues that specification-code equivalence checking is “unnecessary,” but this critique does **not engage with the methodological role** of this component in our pipeline.

In our framework, the equivalence check is not an optional design choice. It is the *only* mechanism that enables:

- automatic evaluation of generated specifications both syntactically and semantically,
- ensuring no uncaptured code behaviours leading to potential risks, and
- meaningful assessment of intent alignment.

Thank you again for your time in this extra situation.

---

### Meta-Review · Area_Chair_GPHZ · 2026-01-09

**Summary:**

This paper introduces VeriEquivBench, a large-scale benchmark for verifiable code generation in Dafny, together with an Equivalence Score, a ground-truth-free metric that evaluates generated formal specifications by checking bidirectional logical implication between code and specifications using a formal verifier. The work is motivated by the scalability limitations of existing benchmarks that rely on human-authored specifications or unit tests, and proposes an automated pipeline for evaluating both specification quality and code–spec alignment.

Two reviewers view the paper positively, considering the Equivalence Score to be a clever and conceptually significant contribution, and praising the scale and complexity of the benchmark, which is substantially larger than prior datasets. These reviewers also find the engineering effort non-trivial and agree that the benchmark exposes meaningful gaps between current LLM capabilities and the requirements of formally verifiable code generation. Concerns raised by these reviewers primarily relate to initially limited empirical evaluation, clarity of exposition, validation of dataset quality, and justification of auxiliary pipeline components. The authors provided extensive rebuttal responses, including additional experiments on open-source models, detailed failure mode analyses, clarification of the Equivalence Score definition, and expanded validation of the benchmark construction pipeline.

In contrast, one reviewer raises a fundamental objection to the paper’s core methodological premise, arguing that requiring logical equivalence between code and specifications is conceptually flawed because “good” specifications may intentionally capture only a subset of a program’s properties. This reviewer views the equivalence-based correctness notion as misaligned with standard practices in formal verification and, on this basis, assigns a strong reject score. The ensuing discussion reveals a philosophical and definitional disagreement regarding the role of specifications: whether they should be allowed to be incomplete abstractions of code behavior, or whether tight, unambiguous specifications are necessary for scalable automatic verification and evaluation.

Overall, the paper is seen as technically ambitious and well-executed, with a clear and novel evaluation objective, but it remains polarizing due to unresolved disagreement over the appropriateness of the equivalence-based evaluation criterion and the reliance on non-formal components (Spec→NL→LLM-as-a-judge) for intent alignment. The decision therefore hinges less on correctness of implementation and more on whether the proposed definition of specification quality and alignment is considered appropriate and valuable for the intended research community.

**Reviewer Concerns:**

First, concerns about limited empirical evaluation were partially resolved. In response to Reviewer dTTt, the authors added evaluations on multiple open-source models (DeepSeek-R1-671B, CodeLlama-70B, Qwen-2.5-Coder-14B), confirming the difficulty of the benchmark beyond proprietary models. They also added extensive failure mode analyses, with detailed qualitative examples illustrating common sources of verification failure, such as missing intermediate clauses, verifier-specific constraints, and mismatches between tight specifications and intended behavior.

Second, issues regarding clarity and correctness of the Equivalence Score definition were acknowledged and corrected. In response to Reviewer LnLN, the authors fixed misleading descriptions of “uniqueness” and clarified that the metric checks bidirectional implication between code and specifications via the Dafny verifier. This improves technical accuracy and presentation.

Third, concerns about benchmark construction and data quality were addressed with additional validation. The authors provided quantitative evidence for the Python-to-Dafny transformation quality by converting hundreds of LeetCode unit tests and reporting a high pass rate, and they clarified the design and purpose of the TagComp pipeline, including detailed analysis of its low yield and failure cases. These additions improve confidence that the benchmark is non-trivial and that failure cases are understood rather than ignored.

Some concerns remain partially unresolved or open to interpretation, even after the rebuttal.

One such issue is the use of the Spec → NL → LLM-as-a-judge pipeline for intent alignment. While the authors justify this step as necessary to bridge the gap between formal specifications and natural-language queries—and provide qualitative evidence of its effectiveness—the step remains non-formal and is not validated against human judgments. Reviewers differ on whether this undermines the paper’s stated goal of rigorous, automated evaluation or is an acceptable pragmatic compromise given the problem setting.

A core unresolved issue is a fundamental disagreement over the definition and role of specifications in this work.

Reviewer Bjj7 argues that requiring logical equivalence between code and specifications is conceptually flawed, because well-designed specifications may intentionally capture only a subset of program properties relevant to the user’s intent. From this perspective, enforcing equivalence conflates specification completeness with correctness and makes the task unrealistically strict or even ill-posed in cases where user intent is underspecified.

The authors, by contrast, argue that tight and unambiguous specifications are necessary to enable scalable, fully automatic verification and evaluation, and that equivalence is defined with respect to the generated code rather than the original natural-language query. They position the Equivalence Score not as a general definition of program correctness, but as a mechanism to automatically evaluate the quality of generated specifications and prepare code for intent-alignment checking.

This disagreement is conceptual rather than empirical and is not fully resolvable within the scope of the rebuttal.

**Reviewer Scores:**

Reviewer dTTt (Score: 6)

This reviewer viewed the Equivalence Score as potentially novel, and considered the benchmark scale a major strength. The primary weaknesses concerned limited evaluation and lack of failure analysis. The authors directly addressed these points by adding open-source model evaluations, extensive failure mode analysis, and detailed clarifications. As a result, this reviewer’s concerns appear largely resolved. I think this reviewer will maintain the original score.

Reviewer LnLN (Score: 4)

This reviewer acknowledged the Equivalence Score as a significant conceptual contribution and praised the benchmark’s scale and complexity, but raised serious concerns regarding explanation accuracy, reliance on non-formal evaluation components, and insufficient validation of dataset quality. The authors corrected misleading explanations, expanded validation of the Python-to-Dafny transformation, and clarified the necessity and role of the Spec→NL→LLM-judge step. These responses partially mitigate the concerns, though some reservations about rigor and evaluation methodology may remain. The score could plausibly move slightly upward, but a strong shift is uncertain.

Reviewer Bjj7 (Score: 2)

This reviewer’s rejection is based on a fundamental conceptual disagreement regarding the appropriateness of enforcing equivalence between code and specifications. Despite extensive back-and-forth discussion and clarifications, the reviewer maintains that good specifications may be intentionally incomplete and that the paper’s correctness notion is misaligned with standard formal verification practice. While mutual understanding improved over the discussion, the underlying philosophical disagreement remains unresolved. It is therefore unlikely that this reviewer’s score would change.

---

### Decision · Program_Chairs · 2026-01-26

Accept (Poster)